# Lagrangian Perturbation Diffusion Steering: Latent Reinforcement Learning for Generative Policies

**Hikmet Simsir** [1]    **Ozgur S. Oguz** [1]

## Abstract

Behavior cloning with high-capacity generative policies achieves strong imitation performance, but is often limited by demonstration coverage and distribution shift. Direct reinforcement learning fine-tuning can improve performance, but updating large action decoders is frequently unstable and sample inefficient. We propose **Lagrangian Perturbation Diffusion Steering (LP-DS)**, a lightweight adaptation method that improves a frozen generative policy by learning a compact noise-space perturbation before decoding. LP-DS optimizes this perturbation with a Lagrangian trust-region objective, improving downstream value while constraining deviation from the latent prior. Across RoboMimic manipulation, OpenAI Gym locomotion, and Adroit dexterous manipulation benchmarks, LP-DS improves sample efficiency, success, and return while maintaining higher action-space entropy than unconstrained noise-space steering, with return improvements of up to 25% over prior baselines. Additional evaluations with flow-matching backbones, a large vision-language-action model, and physical Franka deployment show that LP-DS is not limited to compact diffusion policies or simulated benchmarks. Project page: https://sites.google.com/view/lp-ds/home.

## 1. Introduction

High-capacity generative policies, including diffusion (Wang et al., 2023) and flow-matching decoders (Black et al., 2025; Park et al., 2025), have emerged as a powerful paradigm for continuous control and robotics because they can represent expressive, multimodal action distributions

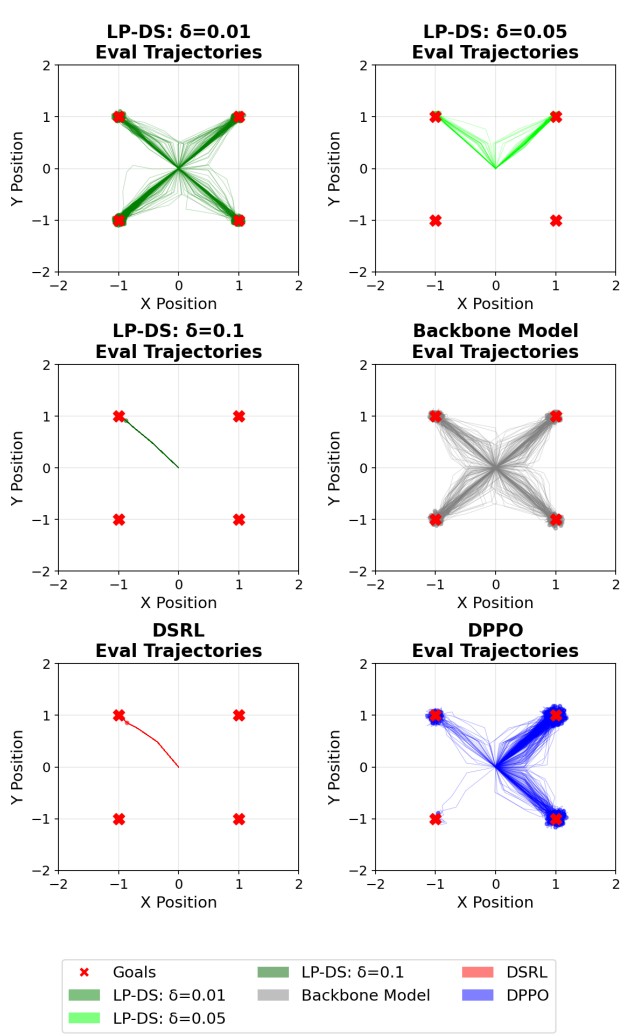

*Figure 1.* **Toy multi-goal navigation with symmetric rewards.** Four equally optimal Gaussian reward peaks (red markers) define four target modes. We visualize evaluation rollouts from the frozen backbone and after adaptation using LP-DS with different trust-region bounds $\delta \in \{0.01, 0.05, 0.1\}$, alongside DSRL (Wagenmaker et al., 2025) and DPPO (Ren et al., 2024). See Sec. 5.2.1 for detailed analysis.

learned from offline data (Ho et al., 2022). Despite strong imitation performance, behavior cloning (BC) policies often inherit limitations of the dataset: demonstrations provide

---

[1]Department of Computer Engineering, Bilkent University, Ankara, Türkiye. Correspondence to: Hikmet Simsir <hikmet.simsir@bilkent.edu.tr>.

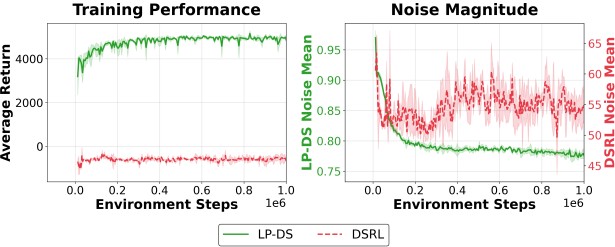

*Figure 2.* **Failure mode of weakly constrained noise-space steering.** Success rate and latent magnitude during online adaptation on HALFCHEETAH-V2 (Brockman et al., 2016), averaged over 3 seeds and using the configuration of Section 5 under the same hard clip ($\|w\| \leq 100$). DSRL predicts higher-magnitude latent queries that correlate with off-manifold decoder behavior and performance degradation, while LP-DS stays closer to the backbone support by design.

incomplete state-space coverage, omit rare recovery behaviors, and can lead to brittle performance under distribution shift (Mehta et al., 2024). Reinforcement learning (RL) (Sutton & Barto, 2018) offers a natural mechanism for online improvement, but directly fine-tuning large generative decoders can be unstable and sample intensive, especially when optimization interacts with long-horizon denoising or integration dynamics (Chandra et al., 2025).

A promising alternative is to improve a frozen generative policy by acting in its latent noise space, treating the decoder as a black-box map from noise to actions. Recent work on *Diffusion Steering via Reinforcement Learning* (DSRL) shows that latent-space RL can efficiently steer diffusion policies without modifying the decoder (Wagenmaker et al., 2025). However, DSRL trains a latent policy that effectively replaces the pre-training prior. This can cause two failure modes: the steered noise can drift away from the standard Gaussian support used to train the decoder, and the learned latent policy can collapse the backbone's multimodal behavior into a narrow subset of modes, as illustrated in Figure 1.

We propose **Lagrangian Perturbation Diffusion Steering (LP-DS)**, a lightweight method for improving frozen generative policies by learning a state-conditioned residual in latent noise space. LP-DS shifts Gaussian noise inputs via $w = \epsilon + \Delta_\theta(s)$ and optimizes $\Delta_\theta$ with a *Lagrangian trust-region objective* that improves downstream value while limiting deviation from the latent prior. This keeps latent queries closer to the backbone's training support while still allowing task-directed policy improvement.

Figure 2 makes this failure mode observable: under weakly constrained noise-space steering, DSRL predicts high-magnitude latent queries that correlate with unstable decoding and performance degradation, whereas LP-DS maintains smaller perturbations through its adaptive trust-region constraint. This distinction is central to LP-DS: rather than replacing the generative prior with an unconstrained latent

policy, it learns a bounded residual shift around the prior.

We evaluate LP-DS across diverse continuous-control and robotics domains and find consistent improvements over representative noise-space steering and diffusion fine-tuning baselines. LP-DS improves success and return while better preserving the behavioral diversity of the frozen backbone, as measured by action-space entropy. We further validate LP-DS across backbone classes, including matched diffusion/flow-matching comparisons, a large vision-language-action $\pi_0$ backbone, and physical Franka robot experiments with simulation-based latent adaptation.

Our main contributions are:

- We introduce **Lagrangian Perturbation Diffusion Steering (LP-DS)**, a lightweight adaptation framework that improves frozen generative policies by learning a state-conditioned residual in latent noise space, avoiding the instability and cost of full decoder fine-tuning.

- We propose a **Lagrangian trust-region objective** that dynamically constrains the perturbation magnitude. The trust-region target $\delta$ provides a controllable mechanism for balancing reward maximization against preservation of the pretrained latent prior.

- We show that LP-DS mitigates mode collapse relative to prior-replacing noise-space steering, preserving higher action-space diversity while achieving strong performance across manipulation, locomotion, and dexterous-control benchmarks.

- We validate LP-DS beyond compact diffusion policies through matched diffusion/flow-matching comparisons, a large vision-language-action $\pi_0$ backbone, and physical Franka robot experiments with simulation-based latent adaptation.

## 2. Related Work

Behavior cloning (BC) is a standard approach for learning robotic control policies directly from demonstrations, and recent progress has been strongly driven by high-capacity generative policy classes. In particular, diffusion-based policies have emerged as an effective and often more stable-to-train backbone for visuomotor control, achieving strong results across manipulation benchmarks (Chi et al., 2024) and broader imitation settings (Pearce et al., 2023). Follow-up work extends diffusion policies toward improved generalization and new domains, including simple 3D representations for scalable visuomotor learning (Ze et al., 2024) and goal-conditioned navigation and exploration (Sridhar et al., 2023). More broadly, recent studies highlight that architectural and training choices can substantially improve the reliability and performance of robotic diffusion transformers (Dasari

et al., 2024), and these trends have begun to carry over into emerging generalist robot foundation models (NVIDIA et al., 2025).

Reinforcement learning (RL) provides principled tools for improving policies beyond demonstration performance, and offline RL in particular has become a major setting for leveraging large static datasets without requiring additional environment interaction (Kostrikov et al., 2021). A growing line of work studies the intersection of offline RL and generative policy classes, where expressive decoders can model complex, multimodal action distributions while value-based objectives guide policy improvement (Wang et al., 2023; Chen et al., 2023; Kang et al., 2023). Beyond standard RL formulations, conditional generative modeling has also been shown to function as a competitive decision-making paradigm on offline benchmarks by directly generating actions conditioned on returns or constraints (Ajay et al., 2023). In robotics, recent work further emphasizes data efficiency in imitation and policy learning pipelines, highlighting complementary directions that reduce demonstration requirements for complex tasks (Ankile et al., 2024). More recently, diffusion policies have been integrated into actor–critic style offline RL via implicit Q-learning, yielding diffusion-parameterized policies that can be improved with stable value learning (Hansen-Estruch et al., 2023). In parallel, online fine-tuning methods that differentiate through diffusion policy rollouts further connect generative policies with policy-gradient optimization (Ren et al., 2024), highlighting both the promise and the practical challenges of adapting high-capacity generative backbones with RL objectives.

A related theme across generative modeling and decision making is to perform optimization directly in the *noise* or *latent* space of a pretrained model, leveraging the strong behavioral prior induced by the generative backbone. In vision, several works improve diffusion outputs by modifying or optimizing the initial noise distribution at inference time, including reward-driven noise optimization for one-step generators (Eyring et al., 2024), noise-dependent initialization and pruning effects in denoising (Mao et al., 2024), and latent-space strategies for generating rare concepts from pretrained diffusion models (Samuel et al., 2023). More broadly, latent-space inference and sampling procedures have been studied as a computationally efficient way to approximate posterior inference and control generation (Venkatraman et al., 2025), while in RL, pretrained behavioral priors similarly serve as strong scaffolds for downstream policy improvement (Singh et al., 2020). Closest to our setting, latent-space reinforcement learning has recently been used to steer diffusion policies by learning a noise-space policy that replaces the base prior during action generation (Wagenmaker et al., 2025). In contrast, LP-DS keeps the pretrained prior as an explicit reference

distribution and learns a *residual* state-conditioned perturbation with an adaptive Lagrangian trust-region constraint, directly targeting stable online improvement while limiting off-manifold latent queries. Finally, recent amortization approaches learn auxiliary networks to predict beneficial noise modifications and reduce test-time compute for diffusion generation (Eyring et al., 2025); unlike these methods, our perturbation module is optimized *with online RL signals* for control and is explicitly regularized to preserve the multimodal structure of a frozen policy backbone rather than accelerating or approximating a fixed generation objective. Broadly, this paradigm of guiding a frozen generative prior with a learned value function parallels high-level planning frameworks like SayCan (Ahn et al., 2022), adapted here to the continuous latent space of low-level control.

## 3. Preliminaries

**Markov decision processes.** We model decision making as a Markov decision process (MDP) $\mathcal{M} = (\mathcal{S}, \mathcal{A}, P, r, \gamma)$, where $\mathcal{S}$ and $\mathcal{A}$ denote the state and action spaces, $P(s' \mid s, a)$ is the transition kernel, $r(s, a)$ is the reward function, and $\gamma \in [0, 1)$ is the discount factor. A policy $\pi(a \mid s)$ induces trajectories via $a_t \sim \pi(\cdot \mid s_t)$ and $s_{t+1} \sim P(\cdot \mid s_t, a_t)$. The reinforcement learning objective is to maximize the expected discounted return $J(\pi) = \mathbb{E}_{\pi, P}[\sum_{t=0}^{\infty} \gamma^t r(s_t, a_t)]$.

The *state–action value function* (Q-function) of a policy $\pi$ is defined as

$$Q^\pi(s, a) := \mathbb{E}_{\pi, P}\left[\sum_{t=0}^{\infty} \gamma^t r(s_t, a_t) \;\middle|\; s_0 = s, \; a_0 = a\right],$$

i.e., the expected discounted return obtained by taking action $a$ in state $s$ and subsequently following policy $\pi$. The corresponding state value function is $V^\pi(s) = \mathbb{E}_{a \sim \pi(\cdot \mid s)}[Q^\pi(s, a)]$.

**Diffusion models.** Diffusion models define a latent-variable generative process via a *forward* noising Markov chain and a learned *reverse* denoising chain (Ho et al., 2020). The forward process incrementally perturbs data with Gaussian noise:

$$\begin{aligned} x_t &= \sqrt{1 - \beta_t}\, x_{t-1} + \sqrt{\beta_t}\, \epsilon_t, \\ \epsilon_t &\sim \mathcal{N}(0, I), \quad t = 1, \dots, T. \end{aligned} \tag{1}$$

This implies the closed-form marginal $x_t = \sqrt{\bar{\alpha}_t}\, x_0 + \sqrt{1 - \bar{\alpha}_t}\, \epsilon$ with $\epsilon \sim \mathcal{N}(0, I)$ and $\bar{\alpha}_t = \prod_{i=1}^{t}(1 - \beta_i)$. Generation runs the reverse-time chain from $x_T \sim \mathcal{N}(0, I)$ using a learned transition kernel

$$\begin{aligned} p_\theta(x_{t-1} \mid x_t) &:= \mathcal{N}\big(x_{t-1}; \mu_\theta(x_t, t), \Sigma_t\big), \\ & t = T, \dots, 1. \end{aligned} \tag{2}$$

Here $\mu_\theta$ is produced by a neural network and $\Sigma_t$ is typically fixed by the noise schedule (Ho et al., 2020). Denoising Diffusion Implicit Models (DDIM) introduces an implicit (optionally deterministic) reverse update that enables fewer sampling steps while preserving the same training objective (Song et al., 2020).

**Flow matching.** Flow matching provides a general ODE-based generative framework by learning a velocity field $v_\theta$ that transports a base noise distribution $p_0$ to the data distribution $p_1$ (Lipman et al., 2023; Liu et al., 2022). The model defines an ODE $dx/dt = v_\theta(x, t)$ for $t \in [0, 1]$, integrated from noise $x(0)$ to data $x(1)$. Training minimizes a regression objective matching $v_\theta$ to the velocity of a conditional probability path $x_t = \psi_t(x_\mathrm{n}, x_\mathrm{d})$:

$$\min_\theta \ \mathbb{E}_{t, x_\mathrm{n}, x_\mathrm{d}}\left[\left\|v_\theta(x_t, t) - \dot{\psi}_t(x_\mathrm{n}, x_\mathrm{d})\right\|_2^2\right], \qquad (3)$$

where $x_\mathrm{n} \sim \mathcal{N}(0, I)$, $x_\mathrm{d}$ is a data sample, and $\dot{\psi}_t$ is the time derivative of the interpolation. Sampling is performed by numerically integrating the ODE from $t = 0$ to $t = 1$.

**Problem setup.** We consider generative policies that predict action chunks using diffusion or flow-matching backbones (Chi et al., 2024; Park et al., 2025). Under deterministic sampling schemes (e.g., DDIM), generation acts as a deterministic mapping from initial noise to output actions (Venkatraman et al., 2025). We therefore model the frozen backbone as a black-box decoder $\Phi : \mathcal{S} \times \mathcal{W} \to \mathcal{A}$, where $\mathcal{W} = \mathbb{R}^d$ denotes the latent noise space (Wagenmaker et al., 2025). Our objective is to improve performance in the MDP $\mathcal{M}$ by adapting the latent input $w$ without updating the potentially heavy decoder $\Phi$.

# 4. Methodology

We present **Lagrangian Perturbation Diffusion Steering (LP-DS)**, a framework for adapting frozen generative policies via residual noise control. We formalize policy improvement as a constrained optimization problem: we seek to maximize downstream value by perturbing the latent input of the generative backbone, subject to a strict trust-region constraint that keeps the modified noise distribution close to the backbone's training prior.

## 4.1. Steering via Residual Perturbation

Let $\Phi : \mathcal{S} \times \mathcal{W} \to \mathcal{A}$ be a frozen, deterministic generative decoder (e.g., a diffusion or flow-matching model) that maps a latent noise vector $w \in \mathcal{W}$ and state $s$ to an action chunk. In the standard setting, actions are sampled by drawing $w$ from a fixed prior, typically $\epsilon \sim \mathcal{N}(0, I)$.

To steer the policy toward higher-reward behaviors without modifying the decoder weights, we parameterize the input

---

**Algorithm 1** Lagrangian Perturbation Diffusion Steering

1: **Init:** $\Delta_\theta(\cdot) \approx 0$, Critics $Q_\psi^{\mathcal{A}}, Q_\phi^{\mathcal{W}}$, $\alpha \geq 0$, Buffer $\mathcal{B}$
2: **for** each env step **do**
3:      **// 1. Data Collection**
4:      Sample $\epsilon \sim \mathcal{N}(0, I)$;    Set $w \leftarrow \epsilon + \Delta_\theta(s)$
5:      $a \leftarrow \Phi(w; s)$; Step env; Store $(s, a, r, s')$ in $\mathcal{B}$
6:      **// 2. Action Critic Update (TD Error)**
7:      Sample batch $B \sim \mathcal{B}$
8:      *// Compute target using next-state learned policy*
9:      Sample $\epsilon' \sim \mathcal{N}(0, I)$;    $w' = \epsilon' + \Delta_{\theta'}(s')$
10:     $a' = \Phi(w'; s')$;    $y = r + \gamma \bar{Q}^{\mathcal{A}}(s', a')$
11:     $\mathcal{L}_\psi = \mathbb{E}_B[(Q_\psi^{\mathcal{A}}(s, a) - y)^2]$;    Update $\psi$ using $\nabla \mathcal{L}_\psi$
12:     **// 3. Latent Critic Distillation**
13:     *// Distill on base noise distribution*
14:     $\mathcal{L}_\phi = \mathbb{E}_{s \sim B, \epsilon}[(Q_\phi^{\mathcal{W}}(s, \epsilon) - Q_\psi^{\mathcal{A}}(s, \Phi(\epsilon; s)))^2]$
15:     Update $\phi$ using $\nabla \mathcal{L}_\phi$
16:     **// 4. Actor & Dual Update (Lagrangian)**
17:     $\mathcal{L}_\theta = \mathbb{E}_{s \sim B, \epsilon}[Q_\phi^{\mathcal{W}}(s, \epsilon + \Delta_\theta(s)) - \alpha(\|\Delta_\theta(s)\|^2 - \delta)]$
18:     $\theta \leftarrow \theta + \eta_\theta \nabla_\theta \mathcal{L}_\theta$
19:     $\alpha \leftarrow [\alpha + \eta_\alpha \mathbb{E}_B(\|\Delta_\theta(s)\|^2 - \delta)]_+$
20:     Update target networks
21: **end for**

---

noise as a state-conditioned residual transformation of the base prior. We define a learnable perturbation network $\Delta_\theta : \mathcal{S} \to \mathcal{W}$ and generate latent queries according to:

$$w = \epsilon + \Delta_\theta(s), \quad \text{where } \epsilon \sim \mathcal{N}(0, I). \qquad (4)$$

This formulation induces a steered policy $\pi_\theta(a|s)$ defined by the pushforward of the base noise distribution through the residual mapping and the frozen decoder $\Phi$. When $\Delta_\theta(s) = 0$, the policy recovers the original behavior cloning distribution exactly.

## 4.2. Constrained Optimization via Lagrangian Relaxation

Our primary objective is to find parameters $\theta$ that maximize the expected return of the steered policy. However, unconstrained optimization in the latent space allows $\Delta_\theta(s)$ to grow arbitrarily large, pushing $w$ off the standard Gaussian manifold. Such off-manifold queries violate the distributional assumptions of the frozen decoder, leading to erratic action generation and mode collapse.

We therefore formulate the learning problem as maximizing the expected Q-value of the generated actions subject to a trust-region constraint. For a given state $s$, let $q_\theta(\cdot \mid s)$ denote the conditional distribution of the steered latent $w = \epsilon + \Delta_\theta(s)$, induced by the base noise $\epsilon \sim p_0 = \mathcal{N}(0, I)$. To ensure the steered noise remains within the valid operating range of the decoder at every encountered state, we constrain the *expected conditional* Kullback–Leibler (KL) divergence

between $q_\theta(\cdot \mid s)$ and the prior $p_0$:

$$\mathbb{E}_{s\sim\mathcal{D}}[D_{\mathrm{KL}}(q_\theta(\cdot \mid s) \,\|\, p_0)]. \qquad (5)$$

Following Eyring et al. (2025), we utilize a tractable approximation of this expected conditional KL divergence for residual perturbations, which is dominated by the magnitude of the shift:

$$\mathbb{E}_{s\sim\mathcal{D}}[D_{\mathrm{KL}}(q_\theta(\cdot \mid s) \,\|\, p_0)] \approx \frac{1}{2}\mathbb{E}_{s\sim\mathcal{D}}\left[\|\Delta_\theta(s)\|_2^2\right]. \quad (6)$$

Based on this approximation, we impose a constraint on the expected perturbation magnitude:

$$\begin{aligned}
\max_\theta \quad & \mathbb{E}_{s\sim\mathcal{D},\epsilon\sim p_0}\left[Q^{\mathcal{W}}(s, \epsilon + \Delta_\theta(s))\right] \\
\text{s.t.} \quad & \mathbb{E}_{s\sim\mathcal{D}}\left[\|\Delta_\theta(s)\|_2^2\right] \leq \delta,
\end{aligned} \qquad (7)$$

where $Q^{\mathcal{W}}$ is a critic estimating the value of latent noise vectors and $\delta > 0$ is a hyperparameter controlling the allowable deviation from the prior.

We solve this constrained problem via the method of Lagrange multipliers. We introduce a dual variable (Lagrange multiplier) $\alpha \geq 0$ and define the Lagrangian $\mathcal{L}(\theta, \alpha)$:

$$\mathcal{L}(\theta, \alpha) = \mathbb{E}\left[Q^{\mathcal{W}}(s, w) - \alpha\left(\|\Delta_\theta(s)\|_2^2 - \delta\right)\right]. \quad (8)$$

We then seek the saddle point of the min-max objective:

$$\min_{\alpha\geq 0}\max_\theta \mathcal{L}(\theta, \alpha). \qquad (9)$$

This formulation naturally creates an adaptive regularization mechanism. If the perturbation violates the trust region ($\mathbb{E}_s[\|\Delta_\theta(s)\|_2^2] > \delta$), $\alpha$ increases, forcing the actor to prioritize the prior constraint over reward maximization. Conversely, if the perturbation is small, $\alpha$ decreases, allowing the actor to explore more aggressive steering.

We alternate between updating the actor $\theta$ and the multiplier $\alpha$. The actor is updated via gradient ascent on Equation 8. The multiplier $\alpha$ is learned via projected dual gradient ascent to enforce the constraint $\alpha \geq 0$:

$$\alpha_{k+1} \leftarrow \left[\alpha_k + \eta_\alpha \mathbb{E}_s\left[\|\Delta_\theta(s)\|_2^2 - \delta\right]\right]_+. \quad (10)$$

### 4.3. Algorithm Summary

The online training procedure is summarized in Algorithm 1. At each step, we collect experience using the steered policy $w = \epsilon + \Delta_\theta(s)$ (Sec. 4.1), update the critics $Q^{\mathcal{A}}$ and $Q^{\mathcal{W}}$ via TD learning and distillation respectively, and finally perform the primal-dual update for $\theta$ and $\alpha$ to maximize reward subject to the trust-region constraint (Sec. 4.2).

## 5. Experiments

**Setup.** We run all methods with the same environment interaction budget and report mean performance across multiple random seeds with $\pm 1$ standard deviation. We evaluate LP-DS on RoboMimic manipulation tasks (Mandlekar et al., 2021), OpenAI Gym locomotion environments (Brockman et al., 2016), and Adroit dexterous manipulation benchmarks (Rajeswaran et al., 2017). We compare **LP-DS** to three representative categories of generative policy improvement: (i) latent noise-space steering (**DSRL**), (ii) policy-gradient fine-tuning (**DPPO**), and (iii) offline-to-online diffusion Q-learning (**IDQL**, **DQL**). Throughout, we treat the generative backbone policy as a frozen module for LP-DS and DSRL. For these cases, we optimize only the adaptation mechanism. We refer the reader to Appendix A.3 for details on the trust-region target $\delta$ (which is set to 0.35 in most experiments) and experiments.

**Backbone Architectures.** Backbone initialization differs by benchmark family. For RoboMimic and Gym, we use the pre-trained diffusion policies released with the DSRL baseline. For Adroit, we train a **flow-matching** generative policy from scratch and use it as the backbone for the steering methods (LP-DS and DSRL), leveraging its efficient deterministic ODE-based sampling. This design choice explicitly demonstrates that LP-DS is *not specific to diffusion models*: the proposed perturbation-and-trust-region framework applies to any generative policy that admits a latent-to-action mapping, including continuous-time flow models. For the fine-tuning baselines (DPPO, IDQL, DQL), we use diffusion policy backbones matched in parameter count and initial performance to ensure fair comparison.

### 5.1. Benchmark Results

Figure 3 reports results across RoboMimic, OpenAI Gym, and Adroit benchmarks averaged over 6 random seeds. Overall, **LP-DS outperforms competing methods on the majority of environments** in terms of success rate or episodic return, while achieving higher sample efficiency and more stable learning dynamics. Across RoboMimic manipulation, LP-DS rapidly reaches high success, especially on precision-sensitive tasks such as SQUARE. In Gym locomotion, LP-DS achieves strong returns with low variance. On WALKER2D, LP-DS reaches approximately 5000 episodic returns compared to roughly 4000 for the strongest baseline, corresponding to a $\sim$25% improvement. In Adroit dexterous manipulation, LP-DS obtains the strongest overall performance across success and reward metrics.

These results suggest that constrained latent steering prevents off-prior adaptation while still allowing effective online improvement. Compared with DSRL, LP-DS avoids uncontrolled latent drift by keeping perturbations close to the prior. Compared with DPPO and diffusion Q-learning

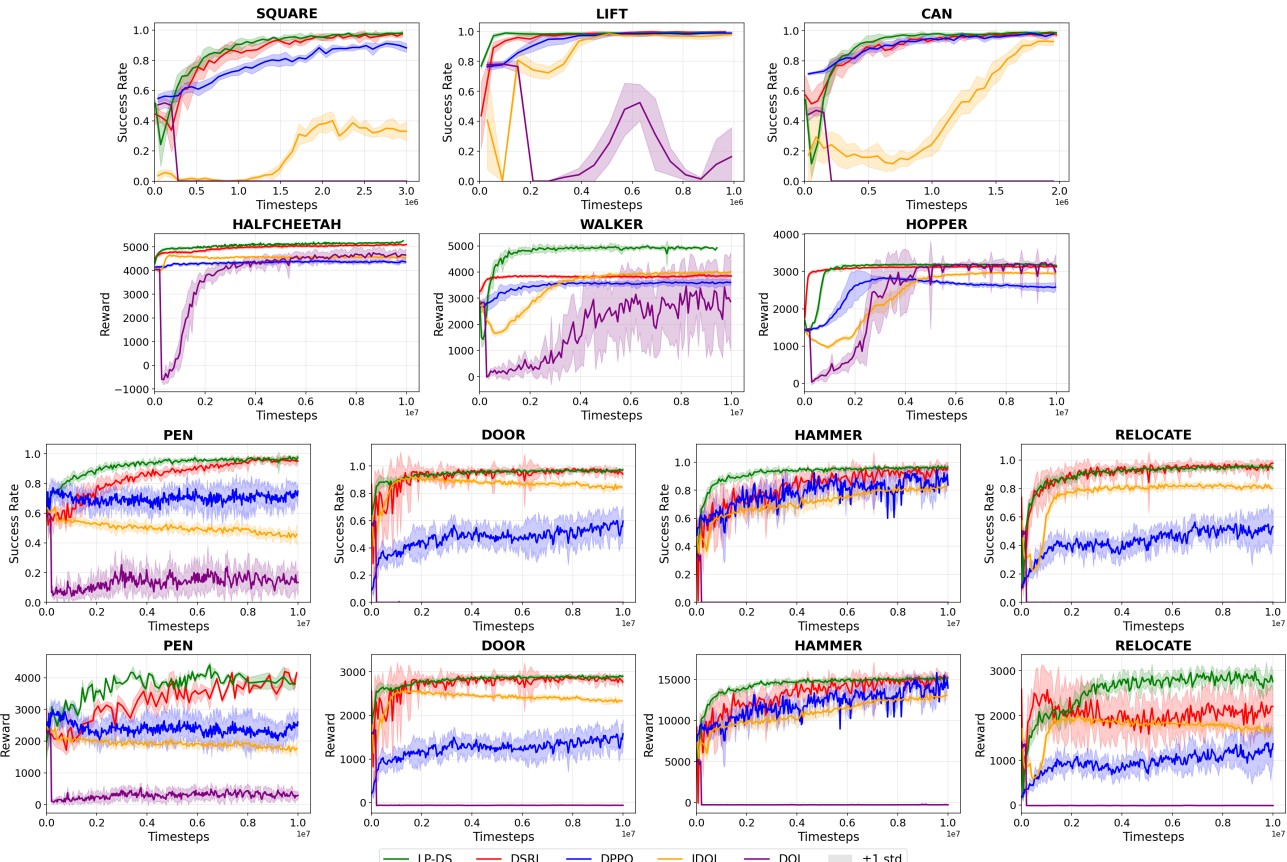

*Figure 3.* **Baseline comparisons across domains. Top row:** RoboMimic manipulation success rates. **Second row:** OpenAI Gym locomotion episodic returns. **Third row:** Adroit dexterous manipulation success rates. **Bottom row:** Adroit dexterous manipulation episodic returns. We compare LP-DS against DSRL (Wagenmaker et al., 2025), DPPO (Ren et al., 2024), IDQL (Hansen-Estruch et al., 2023), and DQL (Wang et al., 2023). Across domains, LP-DS improves performance over baselines.

baselines, LP-DS updates only a lightweight perturbation module rather than fine-tuning the full decoder, improving sample efficiency and stability.

## 5.2. Diversity and Mode Collapse Under Online Adaptation

We analyze how different adaptation strategies affect behavioral diversity during online learning. We study collapse at two complementary levels: (i) *trajectory-level mode coverage* in a controlled toy domain, and (ii) *action-space diversity* in high-dimensional benchmarks using a nonparametric entropy metric.

### 5.2.1. TOY DOMAIN: TRUST-REGION SIZE CONTROLS MODE COLLAPSE

Figure 1 provides a controlled study designed to isolate *mode collapse* and to illustrate the role of the trust-region parameter $\delta$ in LP-DS. The environment contains four Gaussian reward peaks placed symmetrically at equal distance from the origin, yielding four equally optimal target modes.

The frozen backbone policy exhibits multi-goal behavior, reaching all modes across evaluation rollouts. See Appendix B for further explanation and results.

LP-DS exposes a clear and interpretable trade-off between performance and diversity through the choice of $\delta$. With a small trust-region bound ($\delta = 0.01$), LP-DS preserves the multimodal structure of the backbone and consistently reaches multiple goals, albeit with less tightly concentrated trajectories. Increasing the bound to $\delta = 0.05$ yields more directed and higher-quality trajectories while still maintaining coverage of multiple modes. However, when $\delta$ is further increased ($\delta = 0.1$), LP-DS converges to a single dominant mode, exhibiting behavior consistent with mode collapse despite the symmetric reward landscape.

These results demonstrate that $\delta$ acts as an explicit *tuning parameter* controlling the trade-off between maximizing task performance and preserving behavioral diversity. Larger values of $\delta$ allow more aggressive latent-space optimization and faster convergence to high-reward trajectories, but at the cost of collapsing onto a smaller subset of modes. Con-

versely, smaller values of $\delta$ anchor the policy more strongly to the pretrained prior, preserving multimodality while limiting how sharply the policy can specialize.

For comparison, DSRL collapses immediately to a single mode, while DPPO converges to a subset of modes over training. In contrast, LP-DS makes this trade-off *explicit and controllable*, enabling users to balance success rate against mode collapse through a single, interpretable parameter.

### 5.2.2. THE AVOIDING ENVIRONMENT: LONG-HORIZON MULTI-PATH DIVERSITY

To complement the symmetric multi-goal toy domain, we evaluate LP-DS in the AVOIDING environment from the diverse-behavior imitation benchmark of Jia et al. (2024), designed to test long-horizon trajectory-level multimodality. In this task, the agent must reach the goal while navigating around obstacles, and multiple qualitatively distinct routes can solve the task. This setting allows us to study whether online adaptation preserves diverse trajectory modes rather than only maintaining local action-space entropy.

Figure 4 visualizes evaluation trajectories at step 50,000 for LP-DS with different trust-region targets and for DSRL. The results show that the trust-region target $\delta$ again acts as a controllable specialization–diversity dial. With a small trust-region bound ($\delta = 0.01$), LP-DS preserves a broad set of feasible obstacle-avoidance routes, indicating strong trajectory-level multimodality. Increasing the bound to $\delta = 0.05$ yields more directed behavior while still maintaining multiple route choices. With a larger bound ($\delta = 0.3$), the policy concentrates around a narrower successful route, reflecting stronger task specialization. In contrast, DSRL collapses to a single dominant path.

These results reinforce the conclusion from the toy-domain study: LP-DS provides explicit control over the trade-off between multimodal behavior and reward maximization in a longer-horizon sequential task. Quantitative success-rate and reward curves for the AVOIDING environment are provided in Appendix F.

### 5.2.3. ACTION-SPACE DIVERSITY DURING ONLINE ADAPTATION

Figure 5 reports the estimated k-NN action entropy throughout online adaptation, averaged over 3 random seeds. The entropy is computed using the Kozachenko–Leonenko estimator described in Appendix C. Noise-space steering methods (**LP-DS** and **DSRL**) reduce entropy relative to the frozen backbone, as improvement is achieved by concentrating probability mass on higher-value regions of the latent noise space. This concentration translates into reduced diversity after decoding and can be interpreted as latent-space mode selection.

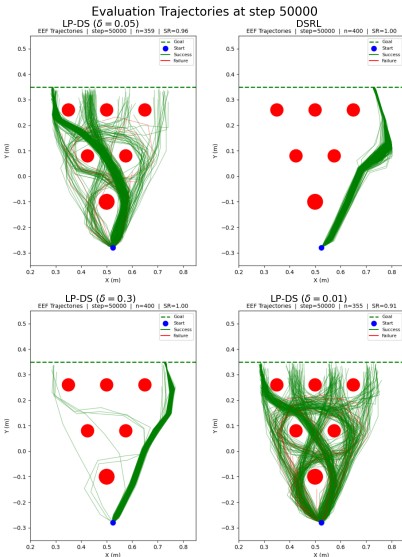

*Figure 4.* **Trajectory-level multimodality in the AVOIDING environment.** We visualize evaluation trajectories at step 50,000 for LP-DS with different trust-region targets and for DSRL. Smaller trust-region targets preserve a broader set of feasible obstacle-avoidance routes, while larger targets produce more concentrated, specialized trajectories. DSRL collapses to a single dominant path.

Importantly, **LP-DS consistently preserves higher action entropy than DSRL** while still improving performance (cf. Figure 3). This gap indicates that the Lagrangian trust-region mechanism in LP-DS mitigates premature collapse by discouraging overly aggressive concentration in the steering space. In contrast, **DPPO** directly fine-tunes the policy in action space and therefore does not exhibit the same systematic entropy decay, remaining closer to the backbone entropy throughout training.

Together with the toy-domain analysis, these results show that LP-DS reduces mode collapse and better utilizes the multimodal capacity of the frozen generative backbone while achieving strong task performance.

### 5.3. Trust-Region Ablations

We study the contribution of the trust-region components in LP-DS by ablating (i) the Lagrangian penalty on the perturbation magnitude and (ii) the explicit bound on the steering noise. Figure 6 reports training success rate (EMA) and estimated action entropy on PEN, comparing **LP-DS**, **LP-DS w/o Lagrangian**, **LP-DS w/o Lagrangian & noise bound**, and **DSRL**.

Removing the Lagrangian penalty leads to unstable learning and degraded success, while removing the noise bound alone does not cause severe performance degradation. As also observed in Figure 2, LP-DS without hard clipping remains competitive across several environments. In con-

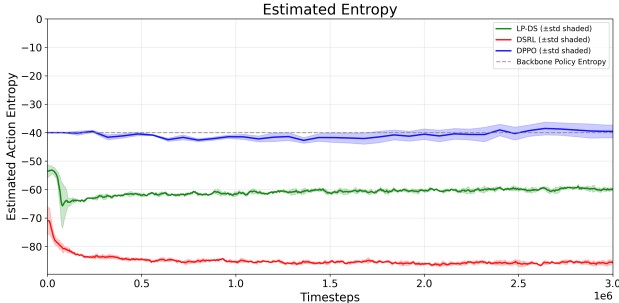

*Figure 5.* **Estimated action entropy during online adaptation (ADROIT PEN).** Results are obtained using the same training and evaluation configuration as the corresponding benchmark runs. We track Kozachenko–Leonenko k-NN action entropy while fine-tuning the same frozen backbone. Noise-space steering methods (LP-DS, DSRL) exhibit an inherent reduction in entropy relative to the backbone, whereas DPPO remains close to the backbone entropy. LP-DS consistently preserves higher entropy than DSRL.

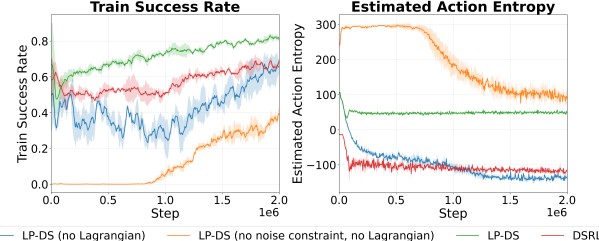

*Figure 6.* **Trust-region ablations on ADROIT PEN.** We plot training success rate (EMA) and Kozachenko–Leonenko k-NN action entropy during online adaptation, using the same training and evaluation configuration as the corresponding benchmark runs and the same frozen backbone across methods. Removing the Lagrangian dual update destabilizes training and reduces final success; removing both the Lagrangian update and the noise bound leads to the worse performance. The full **LP-DS** objective attains the highest success while retaining higher action entropy than the ablated variants and DSRL.

trast, removing both constraints results in highly unstable behavior: **LP-DS w/o Lagrangian & noise bound** frequently explores extreme regions of the steering space that are unlikely under the Gaussian prior used during pretraining, producing low success rates and unreliable learning dynamics. These results indicate that while hard bounding can improve stability, the Lagrangian trust-region constraint is the primary mechanism preventing out-of-distribution steering and preserving meaningful decoding by the frozen generative backbone.

Removing only the *Lagrangian control* yields a different failure mode. **LP-DS w/o Lagrangian** achieves moderate success but exhibits a steady decrease in estimated action entropy over training. This behavior indicates progressive collapse toward a narrow subset of decoded behaviors, consistent with the actor converging to a small region of the steering space (often near a single effective noise realization). In contrast, **LP-DS** maintains substantially higher entropy throughout training while also achieving the strongest success rate, demonstrating that the Lagrangian trust-region objective provides a practical mechanism for simultaneously improving task performance and preserving multimodal behavior.

Finally, **DSRL** exhibits noticeably lower action entropy compared to LP-DS in this setting, reflecting a stronger tendency toward collapse under noise-space steering when deviations are not explicitly regulated. Overall, these ablations support the design choice of adaptive Lagrangian penalty: the Lagrangian update discourages collapse by regulating how aggressively the steering mechanism departs from the base sampling distribution.

## 5.4. Sensitivity to the Trust-Region Target

We next study whether LP-DS requires careful tuning of the trust-region target $\delta$. We sweep $\delta$ across four representative environments: HOPPER, WALKER2D, ROBOMIMIC SQUARE, and ADROIT RELOCATE. Across these environments, we observe that the method is not highly sensitive to the exact value of $\delta$: values above $0.1$ generally yield strong performance, and nearby choices such as $0.35$, $0.5$, and $0.66$ lead to similar reward or success trends. This suggests that $\delta$ acts primarily as a coarse behavioral dial rather than a brittle hyperparameter requiring fine-grained tuning.

Smaller values of $\delta$ enforce conservative steering and better preserve the frozen prior, while larger values permit more aggressive reward maximization. This behavior is consistent with the toy-domain analysis in Section 5.2.1, where $\delta$ controls the trade-off between multimodal preservation and task specialization. Full sweep curves are provided in Appendix E.

## 5.5. Action-Space vs. Noise-Space Perturbation

To test whether optimizing in the latent noise space is necessary, we compare LP-DS against an action-space variant, denoted LP-DS-A, that applies the learned residual directly to the decoded action and optimizes it using the action critic $Q^{\mathcal{A}}$. This ablation removes the latent critic $Q^{\mathcal{W}}$ from the actor update and bypasses optimization over the initial generative noise. As shown in Figure 7, action-space perturbation provides only limited improvement and quickly plateaus around a substantially lower return, whereas noise-space LP-DS continues to improve and reaches much higher final performance. This result supports the role of the latent critic: for high-capacity generative decoders, optimizing the initial noise query is more effective than applying local corrections after decoding.

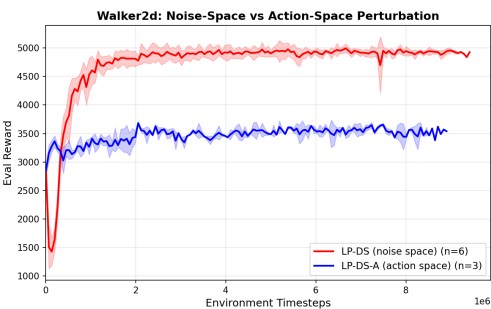

*Figure 7.* **Action-space vs. noise-space perturbation on WALKER2D.** LP-DS optimizes a residual in the latent noise space using the latent critic $Q^{\mathcal{W}}$, while LP-DS-A applies the residual directly in action space using only the action critic $Q^{\mathcal{A}}$. Noise-space perturbation substantially outperforms direct action-space perturbation, indicating that steering the initial latent query is crucial for effectively adapting the frozen generative decoder.

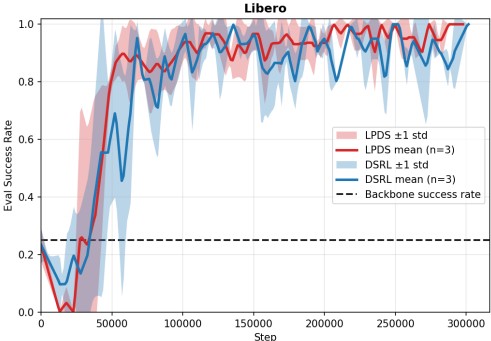

*Figure 8.* **LP-DS on LIBERO with a large VLA backbone.** LP-DS steers a frozen $\pi_0$ vision-language-action backbone using a lightweight perturbation module and improves substantially over the frozen-backbone success rate. The result demonstrates that LP-DS can scale beyond compact generative policies to large Transformer-based VLA models.

### 5.6. Cross-Architecture Robustness

We next evaluate whether LP-DS depends on a specific generative backbone. Beyond compact diffusion and flow-matching policies, we test LP-DS on the LIBERO-90 benchmark "pick up the cream cheese and put it in the tray" task (Liu et al., 2023) using a large vision-language-action backbone. In this setting, the pretrained $\pi_0$ decoder (Black et al., 2025) is kept frozen and only a lightweight perturbation module is trained.

We also conduct a controlled comparison between diffusion and flow-matching backbones on HOPPER-V2 under matched settings. LP-DS achieves comparable final performance with both backbone classes, indicating that the residual perturbation and trust-region formulation are not tied to denoising diffusion chains. The full comparison is provided in Appendix D.

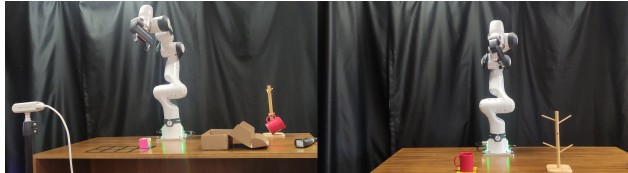

*Figure 9.* **Real-world Franka tasks.** Left: spatial pick-and-place setup, evaluated across a $2 \times 4$ grid of cube initial positions. Right: mug-hanging setup, where the robot must grasp the mug and align its handle with the wooden holder for insertion.

### 5.7. Real-World Robotic Deployment

We additionally evaluate LP-DS on a physical Franka Panda robot to test whether simulation-trained latent steering can transfer to real robot execution. We consider RGBD-conditioned spatial pick-and-place and high-precision mug hanging, shown in Figure 9. In both tasks, the backbone policy is trained from human teleoperation demonstrations and kept frozen; LP-DS performs RL adaptation in a task-matched simulation environment, optimizing only the lightweight latent perturbation module before deploying the resulting steered policy on the physical robot. For pick-and-place, evaluated across a $2 \times 4$ grid with five trials per position, the frozen backbone succeeds in 18/40 trials, while LP-DS succeeds in 33/40 trials. For mug hanging, the frozen backbone succeeds in 11/20 trials, while LP-DS succeeds in 17/20 trials. These results provide initial physical-robot evidence that constrained latent steering can improve frozen generative policies beyond the simulation environment used for adaptation; details are provided in Appendix G.

## 6. Conclusion

We presented **Lagrangian Perturbation Diffusion Steering (LP-DS)**, a lightweight online adaptation method for frozen generative policies. LP-DS improves performance by learning a state-conditioned residual perturbation of the latent prior, steering action generation without modifying the underlying diffusion or flow-matching decoder. The method uses a primal–dual Lagrangian trust-region objective to regulate perturbation magnitude and prevent off-prior latent queries.

Across RoboMimic, OpenAI Gym, Adroit, LIBERO, and physical Franka experiments, LP-DS improves success rates and returns over representative baselines while preserving more behavioral diversity than unconstrained noise-space steering. The trust-region target $\delta$ provides an interpretable control knob for balancing reward maximization and prior preservation. Future work includes adaptive trust-region targets, tighter links between latent constraints and trajectory-level objectives, and broader deployment in partially observable and large-scale vision-language-action settings.

## Impact Statement

This paper presents a method for improving pretrained generative control policies through constrained latent-space adaptation, with the goal of advancing reinforcement learning and generative modeling for robotics and continuous control. By enabling sample-efficient improvement while keeping the generative decoder frozen, LP-DS may reduce the cost of adapting large policies and help preserve the behavioral structure of pretrained robot-control models.

As with other learning-based control methods, unsafe deployment in physical systems could lead to unintended behavior. LP-DS partially mitigates this risk by constraining latent perturbations through a trust-region mechanism, but practical deployment still requires appropriate safety checks, monitoring, and task-specific validation. We do not foresee societal concerns beyond those commonly associated with reinforcement learning, robotics, and generative control.

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

# A. Additional Experimental Details

## A.1. Common Experimental Settings

For fair comparison, we adopt the same environments, evaluation protocols, network architectures, optimizer settings, and training schedules as the DSRL baseline. Unless otherwise specified, all hyperparameters for RoboMimic and OpenAI Gym experiments are identical to those reported in Appendix C.1 of the DSRL paper (Wagenmaker et al., 2025). Implementation given by DSRL is used to run the experiments. This includes backbone architectures, critic designs, replay buffer sizes, update-to-data ratios, and evaluation cadence.

For the baselines DPPO, IDQL, DQL, we used the hyperparameters and implementation provided by (Ren et al., 2024). These parameters are for OpenAI Gym locomotion and Robomimic environments. Please refer to section E10 of the related paper for further details.

LP-DS introduces a single additional hyperparameter beyond DSRL: the trust-region target $\delta$, which controls the allowable magnitude of the latent perturbation. All other settings are inherited unchanged from the DSRL configuration.

## A.2. Adroit Experiments

Table 1 reports the environment-specific hyperparameters used in our Adroit experiments. These values are used for LP-DS and DSRL experiments. Critic gradient steps per update is represented by $Q^W$. We set target entropy of DSRL to 0 on all Adroit tasks.

To train new diffusion policies, we used the code provided by DPPO (Ren et al., 2024). Flow matching policies are trained from scratch.

For DPPO, IDQL, DQL, we used the parameters specified in Table 4.

The same action chunk size ($T_a = 8$) and prediction horizon ($T_p = 1$) were used in all runs.

Table 3 shows the clipping values used in algorithms LP-DS and DSRL in adroit experiments.

## A.3. Trust-Region Targets Across Environments

LP-DS introduces a single additional hyperparameter relative to DSRL: the trust-region target $\delta$. For each environment, $\delta$ is fixed throughout training and selected to balance performance and behavioral diversity, as analyzed in Section 5.2.1.

Table 2 summarizes the trust-region targets used for all benchmark families.

*Table 1.* Hyperparameters for Adroit dexterous manipulation experiments used for LP-DS and DSRL

| Hyperparameter | PEN/HAMMER/DOOR/RELOCATE |
|---|---|
| Hidden size | 256 |
| Gradient steps per update (UTD) | 10 |
| $Q^W$ update steps | 10 |
| Discount factor ($\gamma$) | 0.99 |
| Action magnitude bound | 2.5 |
| Initial environment steps | 2,000 |
| Backbone sampling | ODE |
| Number of actor/critic layers | 3 |
| Number of critics | 2 |

## A.4. Evaluation Protocol

All results are averaged over multiple random seeds (6 for main benchmarks, 3 for entropy analyses and ablations). We use identical seeds across methods to reduce variance. Evaluation rollouts are performed using deterministic decoding for diffusion policies (DDIM) and deterministic ODE integration for flow-matching backbones.

Performance is reported as success rate for sparse-reward tasks and episodic return for dense-reward tasks, following standard benchmark conventions.

Action chunking is considered while calculating the step counts so that for all implementations, one step corresponds to the execution of one action.

# B. Toy Experiment

We design a symmetric multi-goal navigation task to isolate trajectory-level mode collapse under online adaptation. The environment is a 2D continuous-control MDP with state $s = (x, y) \in [-2, 2]^2$ and action $a = (\Delta x, \Delta y) \in [-1, 1]^2$. Transitions follow $s_{t+1} = \text{clip}(s_t + 0.5 \, \text{clip}(a_t, -1, 1), -2, 2)$ with a horizon of 20 steps. The reward is shaped by the distance to the nearest goal, with a sparse bonus upon entering a goal region: $r_t = -\min_{g \in \mathcal{G}} \|s_t - g\|_2 + 10 \, \mathbb{I}[\min_{g \in \mathcal{G}} \|s_t - g\|_2 < 0.2]$, where $\mathcal{G} = \{(\pm 1, \pm 1)\}$ are four corner goals.

To obtain a multimodal behavioral prior, we generate an offline dataset of state–action pairs using a noisy expert rather than interacting with the environment. We sample states uniformly, $s \sim \text{Unif}([-2, 2]^2)$, choose the nearest goal $g^\star(s) = \arg\min_{g \in \mathcal{G}} \|s - g\|_2$, and define the ideal direction action $a^\star(s) = \text{clip}(g^\star(s) - s, -1, 1)$. To model imperfect demonstrations and induce stochasticity, we add i.i.d. Gaussian noise to the expert action and re-clip: $a = \text{clip}(a^\star(s) + \xi, -1, 1)$ with $\xi \sim \mathcal{N}(0, \sigma^2 I)$. Unless otherwise stated, we use $\sigma = 0.1$ and generate $N = 10^6$ samples, yielding a static dataset $\mathcal{D}_{\text{off}} = \{(s_i, a_i)\}_{i=1}^N$.

We then train a diffusion policy backbone from scratch on $\mathcal{D}_{\text{off}}$ via behavior cloning and freeze the decoder for all sub-

*Table 2.* Trust-region target $\delta$ used by LP-DS across environments.

| Environment | Trust-region target $\delta$ |
|---|---|
| RoboMimic Square | 0.35 |
| RoboMimic Lift | 0.10 |
| RoboMimic Can | 0.35 |
| Hopper-v2 | 0.50 |
| Walker2D-v2 | 0.35 |
| HalfCheetah-v2 | 0.35 |
| Adroit Pen | 0.35 |
| Adroit Hammer | 0.10 |
| Adroit Door | 0.35 |
| Adroit Relocate | 0.35 |

*Table 3.* Hard clipping values ($b_W$) used in Adroit environments

| Environment | Clipping value |
|---|---|
| Adroit Pen | 2.5 |
| Adroit Hammer | 2.5 |
| Adroit Door | 2.5 |
| Adroit Relocate | 2.5 |

*Table 4.* Hyperparameter settings for DPPO, DQL and IDQL used in the Adroit environment.

| Method | Hyperparameter | Value |
|---|---|---|
| **DPPO** | $\gamma_{\text{denoise}}$ | 0.99 |
| | GAE $\lambda$ | 0.95 |
| | $N_\theta$ | 5 |
| | $N_\phi$ | 5 |
| | $\epsilon$ | 0.01 |
| | Batch size | 50,000 |
| | $K'$ | 10 |
| **DQL** | $\alpha_{\text{DQL}}$ | 1 |
| | $N_\theta$ | 16 |
| | $N_\phi$ | 16 |
| | Replay buffer size | 1,000,000 |
| | Batch size | 500 |
| **IDQL** | $M_{\text{IDQL}}$ | 20 |
| | $N_\theta$ | 128 |
| | $N_\phi$ | 128 |
| | Replay buffer size | 1,000,000 |
| | Batch size | 1,000 |

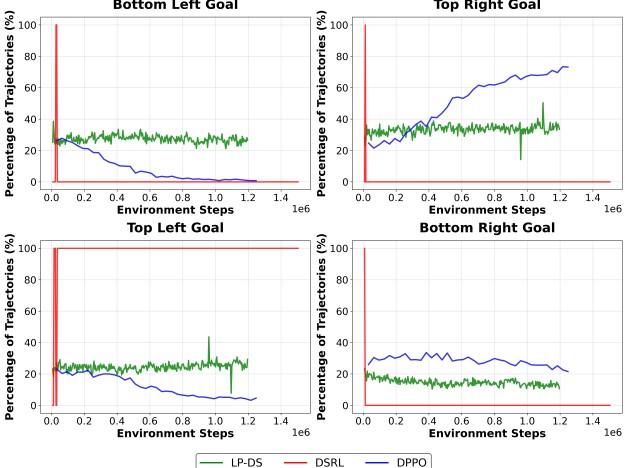

*Figure 10.* **Trajectory-level mode coverage in the symmetric multi-goal toy task.** Each panel corresponds to one goal (bottom-left, top-right, top-left, bottom-right) and shows, as training proceeds, the fraction of evaluation trajectories (out of 1000 rollouts per evaluation) that reach that goal. Concentration of mass into a single panel indicates mode collapse, while sustained non-trivial mass across multiple panels indicates preserved multimodality.

sequent online adaptation methods. Starting from the same frozen backbone initialization, we adapt the policy using LP-DS, DSRL, and DPPO, and study whether adaptation preserves the backbone's multi-goal coverage or collapses to a single goal.

Figure 10 reports trajectory-level mode coverage during training. At each evaluation point, we run 1000 rollouts

and assign each trajectory to one of the four goals based on the goal it reaches (e.g., the first goal region entered, or equivalently the closest goal at termination). Each subplot corresponds to one goal, and the curve value is the fraction of evaluation trajectories reaching that goal. LP-DS is run with parameter $\delta = .01$. A mode-preserving policy maintains non-trivial mass across multiple panels, while mode collapse appears as concentration of probability mass into a single panel. This visualization makes collapse directly observable at the trajectory level, complementing the action-space entropy analysis in Section C.

We used $b_W = 1.5$ as the hard clipping bound for DSRL for this experiment.

## C. Kozachenko–Leonenko Action Entropy Estimator

To quantify behavioral diversity in decoded action space, we estimate the conditional differential entropy of the policy-induced action distribution $p_\pi(a \mid s)$ using the Kozachenko–Leonenko $k$-nearest-neighbor estimator (Kozachenko & Leonenko, 1987). For probe states $\{s_b\}_{b=1}^{B}$ sampled from the replay buffer, we draw $K$ stochastic decoded action samples from the generative policy at each state, yielding $\{a_{b,i}\}_{i=1}^{K}$. Each action sample is flattened across the action-chunk dimensions before entropy estimation.

We compute the estimator independently for each probe

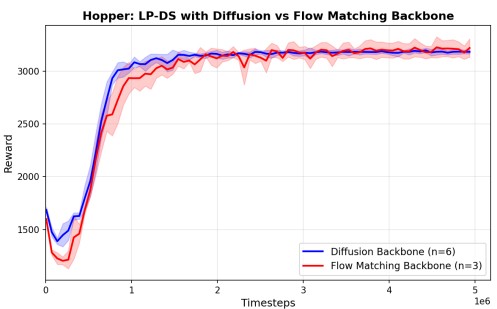

*Figure 11.* **Diffusion vs. flow-matching backbones on HOPPER-v2.** LP-DS achieves comparable final performance when applied to either a diffusion backbone or a flow-matching backbone under matched settings. This indicates that the residual perturbation and trust-region formulation are not tied to denoising diffusion chains.

state and then average across probe states:

$$\widehat{H}_{\mathrm{KL}}(A \mid S) = \frac{1}{B} \sum_{b=1}^{B} \widehat{H}_{\mathrm{KL}}\big(\{a_{b,i}\}_{i=1}^{K}\big), \quad (11)$$

$$\widehat{H}_{\mathrm{KL}}\big(\{a_i\}_{i=1}^{K}\big) = -\psi(k) + \psi(K) + \log c_D$$
$$+ \frac{D}{K} \sum_{i=1}^{K} \log\big(\rho_i^{(k)} + \epsilon\big). \quad (12)$$

Here $D$ is the flattened action dimensionality, $c_D$ is the volume of the $D$-dimensional unit ball, $\psi(\cdot)$ is the digamma function, and $\rho_i^{(k)}$ is the Euclidean distance from sample $a_i$ to its $k$-th nearest neighbor. We add a small numerical constant $\epsilon$ for stability. A decrease in this estimate indicates that the decoded action distribution has become more concentrated around a smaller set of behaviors.

## D. Diffusion vs. Flow-Matching Backbone Comparison

To isolate whether LP-DS depends on a particular generative-policy architecture, we compare diffusion and flow-matching backbones in the same HOPPER-V2 environment under matched settings. Figure 11 shows that LP-DS achieves comparable final performance with both backbone classes. The flow-matching backbone initially learns slightly more slowly, but converges to the same performance range as the diffusion backbone. This supports the claim that LP-DS is not specific to denoising diffusion policies and can also steer ODE-based generative decoders.

## E. Trust-Region Target Sensitivity

LP-DS introduces the trust-region target $\delta$, which controls the allowable magnitude of the learned latent perturbation. In the main experiments, we use a fixed environment-specific value of $\delta$ as listed in Table 2. Here, we study how sensitive the method is to this choice by sweeping $\delta$ across

representative environments and measuring the resulting reward or success trends.

Figure 12 shows the effect of different trust-region targets. Overall, LP-DS is not highly sensitive to the exact value of $\delta$ over a broad range. Very small values impose a conservative trust region, which strongly anchors the policy to the frozen backbone and can limit reward improvement. Larger values allow more aggressive latent-space steering and typically improve task performance, but excessively loose constraints may reduce the regularizing effect of the prior and increase the risk of mode collapse or unstable adaptation. Across the tested environments, moderate values such as $\delta = 0.35$ provide a reliable default, while nearby values often yield similar performance.

These results support the interpretation of $\delta$ as a coarse behavioral dial rather than a brittle hyperparameter. Smaller values favor prior preservation and behavioral diversity, whereas larger values favor task specialization and reward maximization. This trend is consistent with the toy-domain analysis in Section 5.2.1, where increasing $\delta$ shifts LP-DS from conservative multimodal behavior toward more concentrated high-reward behavior.

## F. Avoiding Environment

Figure 13 reports quantitative success-rate and reward curves for the AVOIDING environment from Jia et al. (2024). These curves are consistent with the trajectory visualizations in Section 5.2.2: larger trust-region targets improve specialization and final performance, while smaller targets preserve more diverse behavior.

## G. Real-World Robot Experiments

We evaluate LP-DS on a physical Franka Panda robot using frozen generative backbones trained from human teleoperation demonstrations. The goal of these experiments is to test whether latent perturbation steering learned in simulation can transfer to real robot execution under sensing noise, calibration error, and actuation uncertainty. We consider two physical manipulation tasks, shown in Figure 9: spatial pick-and-place and mug hanging. In the spatial pick-and-place task, the robot must grasp a cube and place it at a target location. In the mug-hanging task, the robot must grasp a mug and hang it by its handle on a stationary holder. Mug hanging is more precision-sensitive because success requires accurate handle alignment and narrow-tolerance insertion.

For each task, we first train a generative backbone policy from human teleoperation demonstrations. We make the generative model condition on the robot joint state, observation received from Intel RealSense Depth Camera 435i

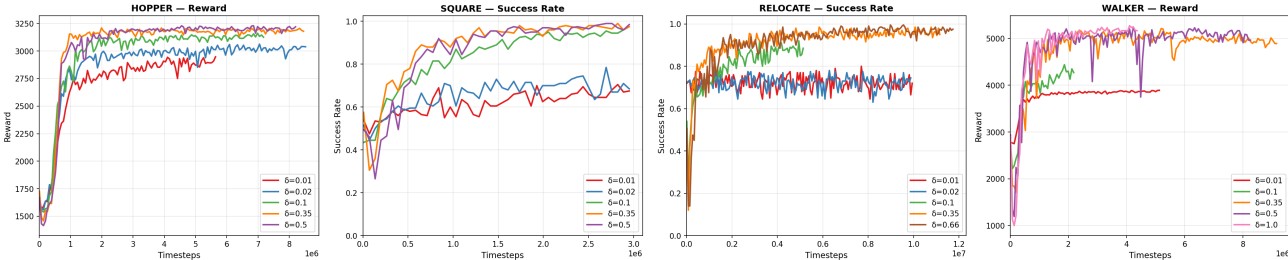

*Figure 12.* **Sensitivity to the trust-region target** $\delta$. We sweep $\delta$ across representative environments and report the resulting reward or success trends. Very small trust-region targets can overly restrict latent steering, while moderate values provide strong performance without requiring fine-grained tuning. The results indicate that $\delta$ acts as a coarse control knob for the trade-off between prior preservation and reward maximization.

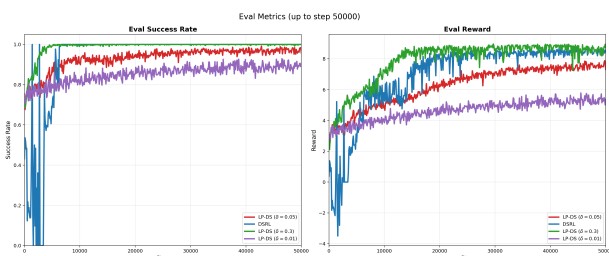

*Figure 13.* **Success rate and reward in the AVOIDING environment.** We report evaluation success rate and evaluation reward for LP-DS with different trust-region targets and for DSRL. The results are consistent with the trajectory visualizations: smaller trust-region targets yield more conservative but diverse behavior, while larger targets produce stronger specialization and higher final task performance.

(only for pick and place task), and the gripper position. The backbone maps the current observation to an action chunk through a flow-matching decoder. After behavior-cloning pretraining, this decoder is kept frozen throughout LP-DS adaptation and physical deployment. For each physical task, we construct a corresponding task environment in the RAI/Robotic simulation framework (Toussaint, 2026). LP-DS adaptation is then performed in this simulated environment rather than directly on the physical robot. During adaptation, only the lightweight latent perturbation module is optimized; the generative decoder remains fixed. The resulting steered policy is then deployed on the physical Franka robot without further fine-tuning of the decoder.

For the spatial pick-and-place task, we evaluate the deployed policy across a $2 \times 4$ grid of object initial positions in the physical robot workspace. Each grid location is evaluated with five independent trials, resulting in 40 total physical trials. Table 5 reports the per-position success rates before and after LP-DS adaptation, using the first five trials from each grid location. Position 1 corresponds to the top-right cell of the workspace grid and position 8 corresponds to the bottom-left cell. The frozen backbone succeeds in 18/40 trials, while LP-DS succeeds in 33/40 trials. For the mug-

*Table 5.* **Per-position success rates for real-world spatial pick-and-place.** Each cell corresponds to one physical object initial position in the $2 \times 4$ workspace grid and reports frozen-backbone success versus LP-DS success over five trials. The table layout matches the physical grid: the top row is positions $7, 5, 3, 1$ and the bottom row is positions $8, 6, 4, 2$.

| Pos. 7 | Pos. 5 | Pos. 3 | Pos. 1 |
|---|---|---|---|
| Frozen: 0/5 | Frozen: 3/5 | Frozen: 3/5 | Frozen: 1/5 |
| LP-DS: 3/5 | LP-DS: 4/5 | LP-DS: 4/5 | LP-DS: 4/5 |
| **Pos. 8** | **Pos. 6** | **Pos. 4** | **Pos. 2** |
| Frozen: 1/5 | Frozen: 3/5 | Frozen: 4/5 | Frozen: 3/5 |
| LP-DS: 4/5 | LP-DS: 5/5 | LP-DS: 4/5 | LP-DS: 5/5 |

*Table 6.* **Real-world Franka evaluation.** LP-DS improves the frozen generative backbone after simulation-based latent adaptation and physical deployment.

| Task | Frozen backbone | LP-DS |
|---|---|---|
| Spatial pick-and-place | 18/40 | 33/40 |
| Mug hanging | 11/20 | 17/20 |

hanging task, we evaluate 20 independent physical trials. The frozen backbone succeeds in 11/20 trials, while LP-DS succeeds in 17/20 trials. Overall physical robot results are summarized in Table 6.

These results provide initial evidence that simulation-trained latent steering can improve frozen generative policies when transferred to real robot execution. Since LP-DS modifies only the latent input to the frozen decoder, the deployed policy remains anchored to the pretrained behavioral prior while still benefiting from task-directed adaptation. Failures are mainly caused by grasp misalignment, visual localization error, or small execution errors near contact-rich parts of the task. Videos are provided on the project page.

