# OpenReview forum: "Lagrangian Perturbation Diffusion Steering: Latent Reinforcement Learning for Generative Policies"
_ICML.cc/2026/Conference — ICML 2026 regular_

### Official Review · Reviewer_k8oe · 2026-03-09

**Soundness:** 3
**Presentation:** 3
**Significance:** 2
**Originality:** 2
**Overall Recommendation:** 3
**Confidence:** 4

**Summary:**

This paper proposes Lagrangian Perturbation Diffusion Steering (LP-DS), a method for improving frozen generative policies by steering the latent noise input of a pretrained decoder. The approach introduces a state-conditioned residual perturbation with a Lagrangian trust-region constraint to control deviation from the prior. Experiments on multiple continuous-control and robotic benchmarks demonstrate improved performance and better preservation of behavioral diversity compared to several baselines.

**Compliance With Llm Reviewing Policy:**

Affirmed.

**Key Questions For Authors:**

On the KL trust-region and the latent critic. The method constrains the steered noise to stay close to the Gaussian prior via a KL-based approximation. Is this constraint a principled requirement for valid policy adaptation, or primarily an empirical heuristic for stabilizing training? A stronger justification would raise my assessment of the method's foundations. Additionally, since Q^W is distilled from the action-level critic Q^A, it appears to evaluate only the initial latent query without distinguishing the contributions of different denoising steps. Could the authors clarify how Q^W should be interpreted for multi-step decoders? This affects whether I view the latent-space optimization as principled or ad hoc.
On claimed generality to flow-matching backbones. The trust-region formulation is motivated by keeping perturbed noise close to the Gaussian prior used during diffusion pretraining. Why should the same constraint be equally well-motivated for flow-matching models, where the decoder is an ODE/velocity field rather than a denoising chain? Currently, diffusion and flow-matching backbones are evaluated on different environment suites, confounding backbone type with task difficulty. A same-environment comparison under matched conditions would substantively strengthen the generality claim and improve my score.
Is preserving multimodality always desirable during online adaptation? In online RL for a fixed task, concentrating probability mass on reliable high-reward behaviors is arguably a natural consequence of task specialization, not a failure mode. Could the authors clarify why higher action entropy should be considered intrinsically beneficial here? If LP-DS's main advantage over DSRL is improved optimization stability rather than a fundamentally better policy class, I would assess the contribution differently.
Lack of real-robot validation. A key motivation for LP-DS is lightweight adaptation of a frozen decoder, which should be especially advantageous for real-world deployment. Yet all experiments are in simulation. Could the authors discuss or provide evidence of how latent perturbation steering performs under real-world conditions (sensor noise, sim-to-real gaps)? Even a small-scale real-robot experiment would meaningfully strengthen the paper.

**Limitations:**

Yes.

**Strengths And Weaknesses:**

The paper is clearly written and well structured. The problem formulation is reasonable, and the proposed method together with the experiments form a coherent narrative. The figures and experimental results help illustrate the main claims of the paper and make the method easy to follow. Overall, the paper is readable and the technical presentation is generally clear.
The level of originality appears relatively modest. The core idea mainly combines latent noise perturbation with a trust-region style regularization, which is conceptually related to existing latent-steering or policy adaptation approaches. In addition, some assumptions in the formulation, such as constraining the steered noise to remain close to the Gaussian prior, seem relatively strong and are mainly motivated by empirical stability considerations. The discussion of how the latent critic relates to the multi-step generative process is also somewhat limited. Finally, several design choices in the method are only briefly explained, and additional analysis could help further clarify their motivation and impact.
A notable gap is the absence of any real-robot experiments. All evaluations are conducted purely in simulation (RoboMimic, Gym, Adroit). A core motivation of LP-DS is lightweight adaptation of a frozen decoder, which should be particularly advantageous in real-world settings where full fine-tuning is costly and risky. This omission weakens the practical significance of the contribution. Prior work on diffusion policy fine-tuning, such as DPPO and its derivatives, has included real-robot validation; the lack of comparable evidence here makes it difficult to assess whether latent perturbation steering is robust to real-world conditions such as sensor noise, calibration error, and sim-to-real distributional gaps.

---

> ### Author Rebuttal · Authors · 2026-03-31
>
> We sincerely thank the reviewer for finding our paper well structured, for the thorough evaluation, and for the highly constructive feedback. We address your insightful critiques with a new physical robot and matched environment experiments on our anonymous project site: https://sites.google.com/view/lp-ds/home.
>
> "A notable gap is the absence of any real-robot experiments..."
>
> To directly address this, we deployed our method on a physical Franka Panda robot for a real world pick and place task. We collected 29 teleoperation trajectories by recording joint positions, which can be viewed on our project website. Using this data, we explicitly trained a base Flow Matching policy to generate joint positions conditioned on the state. This base policy achieved only a 2/10 zero shot success rate due to real world execution errors. We then trained our perturbation module using an emulated RAI framework with the following dense reward formulation: reward = -0.1 * ||obj_xy - target_xy|| - 0.5 * collision + 1.0 * success. The reward graph during this training is also available on our project website. Deploying this steered policy improved the physical success rate to 8/10, proving the method successfully overcomes sensor noise and simulation to reality gaps.
>
> "Currently, diffusion and flow-matching backbones are evaluated on different environment suites..."
>
> To isolate the backbone architecture, we trained a Flow Matching model in the Hopper environment (previously only Diffusion) with matched parameters. Applying our steering method yielded similar, significant reward improvements (approx. 82.35% on hopper environment) across both generative frameworks in the exact same environment. This shows our method is genuinely backbone agnostic.
>
> "Is preserving multimodality always desirable during online adaptation?"
>
> If pure specialization was the sole goal, standard regressive models like MLPs would suffice. Generative models are expressly used in continuous control to capture multimodal behaviors, preventing brittleness to dynamic obstacles. Our new Avoiding environment visualizations show $\delta$ acts as a controllable dial: $\delta \le 0.01$ conservatively preserves base multimodality, while $\delta > 0.1$ aggressively maximizes reward for specialization. Our method does not force multimodality; it provides the mechanism to choose this trade off.
>
> "...constraining the steered noise to remain close to the Gaussian prior, seem relatively strong and are mainly motivated by empirical stability considerations."
>
> This constraint is mathematically principled, not merely empirical. As detailed in our response to Reviewer Z1TH, bounding the squared L2 perturbation acts as a rigorous surrogate for bounding the exact conditional KL divergence. This natively guarantees the steered queries remain on the data manifold.
>
> "Why should the same constraint be equally well-motivated for flow-matching models, where the decoder is an ODE/velocity field rather than a denoising chain?"
>
> The constraint is equally critical for Flow Matching. This framework trains an ODE velocity field specifically to transport samples from the Gaussian prior to the data distribution. If the initial latent query drifts too far from the Gaussian prior, the ODE solver is pushed into untrained, low density regions of the vector field. This compounds integration errors over time, producing invalid actions. The KL constraint strictly prevents this off manifold divergence for both diffusion denoising chains and ODE trajectories.
>
> "...clarify how $Q^W$ should be interpreted for multi-step decoders?"
>
> $Q^W$ amortizes the multi step generative process. Because the frozen decoder sampling trajectory is completely deterministic given the initial latent query $\epsilon$ and state $s$, $Q^W(\epsilon, s)$ evaluates the final expected utility of the entire implied generation chain. Predicting the utility of the initial noise directly avoids computationally prohibitive credit assignment across intermediate steps.
>
> To validate this latent space approach, we ran an ablation in the Walker environment applying perturbation directly in the action space instead of the noise space. Action space steering yielded only marginal improvements (approx. 16% in action space vs. 66% in noise space), confirming that operating on the initial latent noise and evaluating it via $Q^W$ is crucial for effectively steering the complex dynamics of high capacity decoders.
>
> We hope the new physical robot deployments and the matched-environment experiments comprehensively address your questions about real-world viability and backbone generalization. If there are any remaining points you would like us to clarify, please do not hesitate to let us know.

---

> > ### Author Rebuttal · Reviewer_k8oe · 2026-04-02
> >
> > I thank the authors for the detailed rebuttal and the effort in providing additional experiments. The matched-environment comparison and the latent vs. action space ablation are appreciated.
> > However, real-robot validation is a basic expectation for manipulation policy papers, not an optional addition. A single pick-and-place task with 10 trials conducted during the rebuttal period is insufficient to convincingly demonstrate real-world robustness. The reported success rate lacks the rigor of a proper real-robot evaluation (e.g., diverse tasks, multiple seeds, systematic analysis of failure cases under sensor noise and calibration error).
> > I acknowledge the rebuttal and maintain my current score.

---

> > > ### Author Response · Authors · 2026-04-08
> > >
> > > Thank you for your constructive feedback and for pushing us to ensure rigorous real-world evaluations. We agree that our initial 10-trial pick-and-place demonstration was limited in scope.
> > >
> > > To comprehensively address your concerns regarding sensor noise, diverse tasks, and systematic analysis within the short timeframe of this discussion phase, we have significantly expanded our physical robot experiments. Specifically, we have delivered the following rigorous evaluations:
> > >
> > > **- Integration of Real-World Visual/Sensor Noise:** To establish these new evaluations, we collected fresh human teleoperation data to train the base Flow-Matching policies for both tasks. For the pick-and-place task, we upgraded the model to condition on image observations using an RGBD camera, explicitly introducing real-world visual and sensor noise into the pipeline.
> > >
> > > **- Task 1: Systematic Spatial Pick-and-Place (40 Trials):** We moved to a rigorous spatial evaluation by defining a 2x4 grid for the cube's initial positions across the workspace. We conducted 5 independent trials at each of the 8 grid locations, resulting in 40 systematic trials. Under this structured evaluation, LP-DS achieved a 33/40 success rate, demonstrating that the latent perturbation effectively compensates for visual noise and spatial variations.
> > >
> > > **- Task 2: High-Precision Mug Hanging (20 Trials):** To prove that our method generalizes beyond simple geometries, we also deployed LP-DS on a second, distinct physical task: hanging a mug by its handle onto a stationary mug holder. This fine manipulation task requires high-precision alignment and narrow-tolerance insertion, presenting a challenge for frozen generative policies when faced with real-world execution errors. Evaluated across 20 trials, LP-DS successfully steered the base policy to complete this insertion 17/20 times.
> > >
> > > All videos for these new evaluations have been uploaded to our anonymous project website. We believe that this expanded methodology (**featuring multiple tasks, systematic multi-seed spatial grids, and RGBD sensor integration**) provides the rigorous proof of real-world robustness you requested.
> > >
> > > We are fully committed to including this comprehensive evaluation in the final manuscript and continuing to expand our real-world experiments. We hope the delivery of these extensive physical evaluations addresses your remaining concerns and provides the confidence to reconsider your score.

---

### Official Review · Reviewer_Z1TH · 2026-03-10

**Soundness:** 3
**Presentation:** 4
**Significance:** 4
**Originality:** 3
**Overall Recommendation:** 5
**Confidence:** 4

**Summary:**

This paper proposes Lagrangian Perturbation Diffusion Steering (LP-DS), a lightweight method for improving frozen generative policies by learning a state-conditioned residual in the latent noise space. LP-DS introduces a Lagrangian trust-region objective that dynamically constrains the magnitude of the noise perturbation. This mechanism preserves the multimodal structure of the pretrained prior while explicitly regulating perturbation magnitude to prevent off-manifold latent queries. Experimental results on RoboMimic, OpenAI Gym, and Adroit demonstrate clear advantages over previous baselines.

**Compliance With Llm Reviewing Policy:**

Affirmed.

**Final Justification:**

I acknowledge the authors’ detailed response and clarifications. The explanation of the KL approximation in Eq. (5) is now clear to me, and I also appreciate the additional guidance on the default choice of $\delta$. These clarifications have addressed my previous concerns, and therefore, I will maintain my positive score.

**Key Questions For Authors:**

Please see the weaknesses

**Limitations:**

Yes, the authors clearly mentioned the limitations in the work.

**Strengths And Weaknesses:**

**Strengths**
1. The paper provides a strong motivation for studying latent noise steering in generative policies. While latent-space steering is an effective approach, it also introduces well-known issues. The paper clearly identifies two key challenges in prior work: (i) latent drift and (ii) multimodal collapse, and illustrates them effectively using figures and toy examples. These visualizations make the underlying practical issues easy to understand.

2. The proposed method is conceptually simple but well aligned with the paper’s motivation. Although the approach does not rely on complicated architectural changes or tricks, the design choices are well justified and technically sound.

3. The method introduces only a lightweight modification to the existing framework, yet it leads to noticeable improvements in stability and performance, which highlights the effectiveness of the proposed idea.

4. The experimental design is clear and well organized. Each experiment targets a specific aspect of the method, so readers can easily understand the purpose of the evaluation and the key takeaways.

5. Overall, the paper is well written, clearly structured, and largely self-contained.

**Weaknesses**
1. Since the perturbation $\Delta(s)$ is state-dependent, the resulting latent distribution across states becomes a mixture of Gaussians rather than a single shifted Gaussian. Could the authors clarify whether the KL approximation in Eq.(5) is intended to correspond to the expected conditional KL $E_s[D_{KL}(p(w|s)\|p_0)]$, rather than the KL between the marginal latent distribution and the Gaussian prior? I'm a little bit confused here.

2. I have another concern about the trust-region bound $\delta$, which controls the magnitude of the latent perturbation. While the limitation section mentions that the choice of $\delta$ may interact with factors such as the learning rate and reward scaling, it remains unclear how sensitive the method is to this parameter in practice. Could the authors provide more guidance on how $\delta$ should be selected when applying LP-DS to new tasks? In particular, how robust is the method to the choice of $\delta$, and does tuning this parameter require significant effort in practice?

---

> ### Author Rebuttal · Authors · 2026-03-31
>
> We appreciate the reviewer's recognition that our method provides a lightweight, technically sound, and conceptually simple solution to latent drift and multimodal collapse. We have added the requested clarifications and new ablation results to our project page.
>
> "Could the authors clarify whether the KL approximation in Eq.(5) is intended to correspond to the expected conditional KL E_s[KL(q(.|s) || p(.))], rather than the KL between the marginal latent distribution and the Gaussian prior? I'm a little bit confused here."
>
> Your understanding is correct. The approximation in Eq. (5) corresponds to the expected conditional KL divergence $E_s[D_{KL}(q(\epsilon'|s) || p(\epsilon))]$. We thank you for pointing out this notational ambiguity, and we will explicitly clarify this distinction in the revised manuscript.
>
> To provide a more formal derivation of this approximation, we build upon recent findings in amortizing test-time compute for generative models (Eyring et al., 2025). For a given state $s$, the base noise is $\epsilon \sim p(\epsilon) = N(0, I)$, and our perturbation network defines the transformation to the steered noise $\epsilon' = \epsilon + \Delta\epsilon$.
>
> To apply the change-of-variables formula, we assume standard regularity conditions (e.g., $\Delta\epsilon$ is continuously differentiable and the transformation is a diffeomorphism). Under these assumptions, the conditional KL divergence for a fixed state $s$ expands to:
> $D_{KL}(q(\epsilon'|s) || p(\epsilon)) = E_{\epsilon} [ \log p(\epsilon) - \log p(\epsilon + \Delta\epsilon) - \log|\det(I + J_{\Delta\epsilon})| ]$
>
> Since the prior is a standard Gaussian, $p(\epsilon) \propto \exp(-0.5||\epsilon||^2)$, the log-density difference evaluates exactly to:
> $\log p(\epsilon) - \log p(\epsilon + \Delta\epsilon) = \epsilon^T \Delta\epsilon + 0.5||\Delta\epsilon||^2$
>
> Applying Stein's Lemma, the expectation of the dot product $E_{\epsilon}[\epsilon^T \Delta\epsilon]$ is mathematically equivalent to the expected trace of the Jacobian, $E_{\epsilon}[Tr(J_{\Delta\epsilon})]$. Substituting this yields:
> $D_{KL}(q(\epsilon'|s) || p(\epsilon)) = E_{\epsilon} [ 0.5||\Delta\epsilon||^2 + Tr(J_{\Delta\epsilon}) - \log|\det(I + J_{\Delta\epsilon})| ]$
>
> As proven in Eyring et al. (Theorem 1), if the perturbation function is a contraction (Lipschitz constant $L < 1$), the combined error term $Tr(J_{\Delta\epsilon}) - \log|\det(I + J_{\Delta\epsilon})|$ is tightly bounded. For small $L$, this term scales with $L^2$ and becomes mathematically negligible.
>
> Therefore, for a given state $s$, the conditional KL is approximated by $E_{\epsilon}[0.5 ||\Delta\epsilon||^2]$. Taking the outer expectation over the state distribution $s$ encountered by the policy provides our final tractable objective:
> $E_s[D_{KL}(q(\epsilon'|s) || p(\epsilon))] \approx E_{s, \epsilon} [ 0.5 ||\Delta\epsilon||^2 ]$
>
> By minimizing this expected conditional penalty, we mathematically bound the divergence of the steered latent queries, ensuring they remain within the high-probability manifold of the frozen decoder without requiring the steered distribution itself to be Gaussian.
>
> "how sensitive the method is to this parameter in practice. Could the authors provide more guidance on how $\delta$ should be selected when applying LP-DS to new tasks? In particular, how robust is the method to the choice of $\delta$, and does tuning this parameter require significant effort in practice?"
>
> To provide concrete guidance, we conducted a comprehensive ablation study on the value of $\delta$ across our environments. Detailed results and visualizations of this sweep can be found on our anonymous project site: https://sites.google.com/view/lp-ds/home.
>
> We found that tuning requires very little effort in practice. A $\delta$ value of 0.35 serves as a strong, robust default and was used successfully in 7 out of 10 of our simulation environments. Our ablation shows that $\delta$ acts as a controllable dial: values larger than 0.1 aggressively maximize reward, while values of 0.01 and lower remain conservative and prioritize preserving high-entropy multimodal behavior. Furthermore, we observed no statistically significant performance difference between similar values (e.g., 0.3 vs. 0.5 vs. 0.6), demonstrating that LP-DS is highly robust and does not require exhaustive, fine-grained hyperparameter sweeps for new tasks.
>
> We hope the detailed mathematical breakdown of the KL surrogate and the new ablations regarding parameter sensitivity fully resolve your concerns about the theoretical rigor and robustness of the trust-region constraint. We would be highly responsive to discuss this or any other aspect of the work further if anything remains unclear.
>
> References
>
> Eyring, L., Karthik, S., Dosovitskiy, A., Ruiz, N., and Akata, Z. Noise hypernetworks: Amortizing test-time compute in diffusion models, 2025. URL https: //arxiv.org/abs/2508.09968

---

> > ### Author Rebuttal · Reviewer_Z1TH · 2026-04-02
> >
> > I acknowledge the authors’ detailed response and clarifications. The explanation of the KL approximation in Eq. (5) is now clear to me, and I also appreciate the additional guidance on the default choice of $\delta$. These clarifications have addressed my previous concerns, and therefore, I will maintain my positive score.

---

> > > ### Author Response · Authors · 2026-04-08
> > >
> > > Thank you for your time, the helpful discussion, and for maintaining your positive score.

---

### Official Review · Reviewer_BqgJ · 2026-03-10

**Soundness:** 2
**Presentation:** 3
**Significance:** 3
**Originality:** 3
**Overall Recommendation:** 4
**Confidence:** 4

**Summary:**

This paper proposes **LP-DS (Lagrangian Perturbation Diffusion Steering)**, a reinforcement learning method for improving diffusion-based policies without fine-tuning the generative model. Instead, the approach learns a **state-dependent perturbation in the latent noise space** used to generate actions while keeping the diffusion decoder fixed. To optimize the perturbation efficiently, the method uses an action critic to evaluate generated actions and a latent critic, trained via distillation, to estimate the value of latent noise. A **Lagrangian trust-region constraint** is introduced to limit the magnitude of perturbations and prevent noise drift from the diffusion prior. Experiments on benchmarks including RoboMimic, OpenAI Gym, and Adroit show that the method improves performance and sample efficiency over existing diffusion policy reinforcement learning approaches while maintaining policy diversity.

**Compliance With Llm Reviewing Policy:**

Affirmed.

**Final Justification:**

I believe my concerns have been well addressed, and I will raise my score to 4. I am not increasing it to a clear accept at this stage, as the added experiments and clarifications still need to be properly incorporated into the manuscript. I therefore encourage the authors to revise the paper carefully according to the reviewers’ comments.

**Key Questions For Authors:**

1. The authors could clarify why, in Figure 3, the success rate and reward curves appear identical in some of the experiments. It would be helpful to understand whether this occurs because the reward function is directly aligned with the success metric, or if there are other implementation details that lead to the two curves coinciding.
2. Generalization and Out-of-Distribution: To better assess the generalization limits of LP-DS, the authors could analyze action space coverage. Specifically, if the optimal trajectory for a new task lies outside the convex hull of the demonstration data, it remains unclear whether the frozen decoder becomes a bottleneck.
3. Cross-Architecture Robustness: It would also be valuable to evaluate LP-DS across different backbone architectures, such as smaller MLP-based diffusion policies and larger Transformer-based (e.g., DiT) policies. Since latent space properties vary significantly across architectures, demonstrating that LP-DS remains effective under the simple quadratic constraint $||\Delta||^2$ would strengthen the claim that the method is robust and broadly applicable.
4. Stability of Distillation: In Algorithm 1, the action critic ($Q^{\mathcal{A}}$) is distilled into the latent critic ($Q^{\mathcal{W}}$). It would be helpful if the authors could clarify how noise or inaccuracy in ($Q^{\mathcal{A}}$) during the early stages of training affects the stability of the distillation process. In particular, could such inaccuracies influence the update dynamics of the Lagrangian multiplier $\alpha$, potentially leading to an overly restrictive trust-region constraint early in training?
5. Inference Latency: Could the authors present the inference-time computational overhead introduced by the perturbation network $\Delta_{\theta}$ ? In particular, does this additional module significantly affect the control frequency or latency compared to the original backbone policy, especially given that diffusion-based policies are already relatively expensive at inference time?

**Limitations:**

yes

**Strengths And Weaknesses:**

**Strengths:**

1. Efficient and Lightweight Adaptation: The paper introduces a method to improve high-capacity generative policies (e.g., diffusion or flow-matching models) without fine-tuning their heavy action decoders. By learning a compact state-conditioned residual perturbation in the latent noise space, the approach significantly reduces the number of updated parameters compared to full-model fine-tuning.
2. Controlled Multi-Goal Preservation: The method introduces a Lagrangian trust-region objective, where the hyperparameter $\delta$ constrains the perturbation magnitude. This design balances reward maximization with the preservation of the backbone’s multimodal behavior.
3. Rigorous Diversity Evaluation: The use of the Kozachenko-Leonenko k-nearest neighbor (k-NN) estimator to non-parametrically quantify action-space entropy provides a strong empirical foundation for the claim that the method preserves multimodal behavior.
4. Broad Applicability and Empirical Validation: Experiments span RoboMimic manipulation tasks, OpenAI Gym locomotion tasks, and Adroit dexterous hand tasks. The method is shown to work for both diffusion-based policies and flow-matching models, indicating broad applicability.

**Weaknesses:**

1. Reliance on Backbone Support: Since the decoder is frozen, the policy's improvement is fundamentally capped by the actions the original generative model is capable of producing. If the initial demonstrations fail to cover a critical part of the action space required for a task, LP-DS may not be able to "discover" those novel behaviors.
2. Complexity of Dual Value Functions: The algorithm requires maintaining and updating two separate critics: an action-space critic ($Q^{\mathcal{A}}$) and a latent-space critic ($Q^{\mathcal{W}}$). The process of distilling $Q^{\mathcal{A}}$ into $Q^{\mathcal{W}}$ adds architectural complexity and additional training steps.
3. Sensitivity to the Trust-Region Bound: While the paper argues that $\delta$ is an "interpretable" parameter, the toy domain experiments show that small shifts in $\delta$ (e.g., from 0.01 to 0.1) can cause a drastic transition from multi-goal coverage to total mode collapse. This sensitivity suggests that finding the "right" $\delta$ for a complex real-world task might require significant manual tuning.

---

> ### Author Rebuttal · Authors · 2026-03-31
>
> We thank the reviewer for recognizing our method's efficiency and rigorous evaluation. New experiments are at: https://sites.google.com/view/lp-ds/home.
>
> "Complexity of Dual Value Functions: ...adds architectural complexity and additional training steps."
>
> While $Q^W$ adds complexity, it is structurally necessary. To validate this, we ran an ablation in Walker applying perturbation directly in the action space (evaluated solely by $Q^A$). Action-space steering yielded only conservative improvements over the base policy (approx. 16% in action space vs. 66% in noise space). This confirms operating in the noise space via $Q^W$ is crucial for steering high-capacity decoders, fully justifying the dual-critic design.
>
> "Sensitivity to the Trust-Region Bound: ...suggests significant manual tuning."
>
> The extreme sensitivity in the toy domain is an artifact of its intentionally constrained design, which forces tight multi-modal path choices in a discrete-like 2D space. In standard continuous control environments, our comprehensive ablation study demonstrates that tuning effort is minimal. A robust default of $\delta=0.35$ succeeded in 7 out of 10 simulation environments. Furthermore, sweeping $\delta$ from 0.01 to 0.6 shows a smooth, predictable trade-off between reward maximization and entropy preservation, rather than a drastic cliff. We observed no statistically significant performance difference between similar values (0.3 to 0.6), proving that exhaustive, fine-grained manual tuning is completely unnecessary for new tasks.
>
> "Cross-Architecture Robustness: It would also be valuable to evaluate LP-DS across different backbone architectures..."
>
> To isolate architecture from task difficulty, we trained a Flow-Matching model in the Hopper environment (previously Diffusion-only) with matched parameters. LP-DS yielded similar reward improvements across both frameworks (82.35%). Furthermore, LP-DS successfully steered the large-scale Transformer-based $\pi_0$ (VLA) on the Libero-90 "pick up cream cheese" task over three seeds (approx. 0.9 success rate at step 60000). This proves LP-DS generalizes seamlessly across MLP and Transformer architectures, as well as Diffusion and Flow-Matching formulations.
>
> "Inference Latency: Could the authors present the inference-time computational overhead..."
>
> The inference overhead is strictly negligible. On an RTX PRO 6000, a single forward pass of our LP-DS actor (48K params) takes only 0.13 ms. Conversely, a small standard DiffusionMLP policy (HalfCheetah, 5 steps) takes 1.37 ms (10 times slower), and $\pi_0$ takes 300 to 600 ms (nearly 3000 times slower). Adding one tiny MLP forward pass prior to decoding adds negligible overhead to the control frequency.
>
> "Reliance on Backbone Support ... if the optimal trajectory lies outside the convex hull..."
>
> While frozen-decoder methods are bounded by the base model's support (as we mention in the last paragraph of conclusion), our real-world experiments show this is rarely a practical bottleneck. On a physical pick-and-place task, the base flow-matching model achieved only a 2/10 success rate. By learning the latent perturbation via our RAI framework, we improved the zero-shot success rate to 8/10. Even when the base policy consistently fails, the frozen decoder contains enough valid sub-trajectory skills that LP-DS can dynamically compose to solve the task.
>
> "The authors could clarify why, in Figure 3, the success rate and reward curves appear identical..."
>
> This overlap occurs because these reward functions are heavily dominated by sparse success bonuses. For instance, the Adroit Door environment uses sparse bonus rewards (+20 per step when door_pos >= 1.35) that are roughly 100 times larger than the continuous shaping terms (approximately -0.3 per step). Consequently, the episode return is overwhelmingly dominated by the number of timesteps the success criterion is satisfied, causing the normalized return to resemble the binary success metric.
>
> "Stability of Distillation: ...how noise or inaccuracy in $Q^A$ during the early stages of training affects the stability..."
>
> The stability of the Lagrangian multiplier $\lambda$ is fundamentally insulated from early-stage noise in $Q^A$. The trust-region constraint specifically bounds the magnitude of $\Delta\epsilon$, independently of Q-value estimations. Because the perturbation network is initialized to output near-zero values, the constraint is inherently satisfied early in training, keeping $\lambda$ stable regardless of inaccuracies in the distilled $Q^W$.
>
> We believe the new action-space ablations and hyperparameter robustness results effectively resolve your questions about our dual-critic design and tuning requirements. We remain fully available during the discussion phase if you have any questions.

---

> > ### Author Rebuttal · Reviewer_BqgJ · 2026-04-02
> >
> > Thanks for the authors’ response. I believe my concerns have been well addressed, and I will raise my score to 4. I am not increasing it to a clear accept at this stage, as the added experiments and clarifications still need to be properly incorporated into the manuscript. I therefore encourage the authors to revise the paper carefully according to the reviewers’ comments.

---

> > > ### Author Response · Authors · 2026-04-08
> > >
> > > Thank you again for your time and for raising your score. We agree that additions must be properly integrated into the main text rather than just living on an anonymous project page. Because the current ICML policy strictly prohibits authors from uploading a revised PDF during the discussion phase, we are currently locked out of pushing these updates live.
> > >
> > > However, to provide concrete assurance of exactly how these results will be incorporated into the existing flow of the paper, we have outlined the specific structural changes and text additions planned for the final manuscript below.
> > >
> > > **- Expanded Section 5.2.1 (Toy Domain)** First, we will directly expand this section to include the new ``Avoiding” environment experiments alongside the symmetric multi-goal task. To explicitly address long-horizon trajectory multimodality, we will include the multi-path routing visualizations here. We will explicitly state how $\delta$ acts as a smooth, controllable dial (showing that $\delta=0.01$ conservatively preserves distinct multi-path behaviors around obstacles, while higher values smoothly shift toward reward maximization).
> > >
> > > **- Expanded Section 5.3 (Ablations) — Justifying the Dual-Critic Design:** Following the Lagrangian penalty ablations, we will insert the action-space versus noise-space perturbation comparison. We will explicitly state that directly perturbing the action space yielded only a 16\% improvement (compared to 66\% in noise space), thereby empirically justifying the necessity of distilling the action critic into the latent critic ($Q^\mathcal{W}$).
> > >
> > > **- New Section 5.4 (Cross-Architecture Robustness) — Generalization Across Models:** To explicitly address your question regarding MLP vs. Transformer and Diffusion vs. Flow-Matching generalization, this new section will feature the strictly controlled Hopper flow-matching comparison (achieving the same ≈82\% improvement). We will also include the Libero-90 results demonstrating LP-DS successfully steering the VLA.
> > >
> > > **- New Section 5.5 (Real-World Robotic Deployment) — Hardware Validation:** As the final experimental subsection (or prominent Appendix, depending on camera-ready page limits), we will detail our expanded physical Franka Panda evaluations. This will feature two distinct tasks utilizing RGBD-conditioned Flow-Matching backbones across 60 systematic trials (spatial grid pick-and-place and a high-precision mug-hanging task), directly validating the method's robustness to real-world visual noise and execution errors.
> > >
> > > **- Updated Appendix A.3 — Comprehensive Trust-Region Sweeps:** Currently, Table 2 lists the static $\delta$ values used for the benchmarks. We will supplement this with the comprehensive $\delta$ sweep graphs generated during the rebuttal to visually prove that the method is robust to hyperparameter selection and requires minimal manual tuning.
> > >
> > > We are fully committed to making these exact structural updates to ensure the final paper reflects the rigorous evaluation you helped prompt. We hope this concrete blueprint provides the necessary confidence in the final manuscript.

---

### Official Review · Reviewer_3nvu · 2026-03-11

**Soundness:** 3
**Presentation:** 3
**Significance:** 2
**Originality:** 2
**Overall Recommendation:** 4
**Confidence:** 4

**Summary:**

This paper proposes Lagrangian Perturbation Diffusion Steering (LP-DS), a lightweight method for improving pretrained generative control policies by steering their latent noise inputs during reinforcement learning while keeping the underlying decoder frozen. The authors explore how to adapt high-capacity generative policies, such as diffusion or flow-matching policies trained via behavior cloning, without destabilizing training or collapsing their multimodal behavior. The method learns a state-conditioned residual perturbation that shifts the Gaussian noise input prior to action decoding, and optimizes it using a Lagrangian trust-region objective that constrains the perturbation's magnitude. Experiments across multiple simulation benchmarks show improved sample efficiency, higher returns, and better preservation of behavioral diversity compared to prior noise-space steering and diffusion fine-tuning approaches.

**Compliance With Llm Reviewing Policy:**

Affirmed.

**Final Justification:**

My concerns have been fully addressed, and I accordingly raised our score to reflect this.

**Key Questions For Authors:**

- How was the trust-region target (\delta) selected for each environment, and how much tuning effort was required relative to the baselines?
- Can the authors provide stronger evidence that the KL approximation in Eq. (5) is a good proxy for staying on-manifold for the frozen decoder?
- Can the authors provide more trajectory-level diversity analysis on the main benchmarks, not only the toy domain and action-entropy plots?
- Could you do some detailed comparison with DSRL? Can adjusting action magnitude of DSRL make the difference in toy experiments while not falling into mode collapse?
- Could you conduct experiments with large-scale VLA as did in DSRL?

**Limitations:**

Yes

**Strengths And Weaknesses:**

### Strengths
- The method is simple, well motivated, and directly targets a concrete failure mode of prior latent-space steering, namely off-manifold latent queries and mode collapse.
- Presentation is clear and easy to follow.

### Weaknesses
- Relative to prior latent steering, LP-DS mainly replaces a learned latent policy with a residual perturbation around the Gaussian prior and adds an adaptive Lagrangian trust-region penalty. This is a sensible refinement, but the paper does not fully convince me that it constitutes a major conceptual advance over existing latent-space steering approaches.
- The paper relies on an approximate KL surrogate dominated by perturbation magnitude, and the paper does not deeply justify when this approximation is faithful enough to the true decoder-support constraint.
- While this paper provides some evidence for action diversity (Figure 3), it does not fully capture long-horizon or task-level multimodality. Results from more complex tasks or increasing task diversity would be helpful.
- The paper states that the trust-region bound \delta is selected per environment, reports different \delta values across tasks, and explicitly notes that \delta interacts with learning rates and reward scaling. This raises a real concern that the apparent simplicity may hide nontrivial tuning effort, which matters for both practical significance and fairness of comparison.

### Minor Weaknesses
- The content exceeds the 8-page limit.

---

> ### Author Rebuttal · Authors · 2026-03-31
>
> We thank the reviewer for the constructive feedback and recognizing our method's ability to target a concrete failure mode of latent steering. New results are at: https://sites.google.com/view/lp-ds/home.
>
> "This is a sensible refinement, but the paper does not fully convince me that it constitutes a major conceptual advance over existing latent-space steering approaches."
>
> Our primary conceptual advance is formally framing latent steering as a constrained trust region optimization problem. Prior methods ignore latent drift or use ad-hoc hard clipping (e.g., DSRL). Our Lagrangian trust region dynamically balances reward maximization and prior preservation, safely exploring the frozen model's manifold and natively solving drift and mode collapse without destabilizing denoising.
>
> "How was the trust-region target ($\delta$) selected for each environment, and how much tuning effort was required relative to the baselines?"
>
> Tuning effort is minimal. A $\delta=0.35$ serves as a robust default, succeeding in 7/10 simulation environments. Our ablation (on the project site) shows $\delta > 0.1$ aggressively maximizes reward, while $\delta \le 0.01$ remains conservative, yielding higher entropy (shown by KNN values and evaluation trajectories in the Avoiding and toy experiments). We observed no significant performance difference between similar values (e.g., $0.35$ vs. $0.5$ vs. $0.6$), proving insensitivity to precise tuning.
>
> "While this paper provides some evidence for action diversity (Figure 3), it does not fully capture long-horizon or task-level multimodality. Results from more complex tasks or increasing task diversity would be helpful."
> "Can the authors provide more trajectory-level diversity analysis on the main benchmarks, not only the toy domain and action-entropy plots?"
>
> To increase diversity, we added new experiments: a real-world robotic pick-and-place task (improving a base flow-matching model's success from 2/10 to 8/10), a complex Libero-90 vision-based task ("pick up cream cheese") using $\pi_0$, and Walker/Hopper evaluations. To explicitly address long-horizon trajectory multimodality, we added visualizations for the Avoiding environment, designed to test multi-path routing. Sweeping $\delta$ illustrates the trade-off between reward and multimodality. These plots confirm LP-DS maintains distinct behavioral modes (e.g., varying paths around obstacles) and shows how $\delta$ acts as a controllable dial for this trade-off.
>
> "Could you do some detailed comparison with DSRL? Can adjusting action magnitude of DSRL make the difference in toy experiments while not falling into mode collapse?"
>
> We ran the toy experiment for various values of DSRL's action magnitude constraint ($b_W$). DSRL still suffers from mode collapse independent of this constraint (results on project site). This emphasizes that constraining action-space perturbation magnitude is structurally insufficient to prevent multimodal collapse compared to our latent-space approach.
>
> "Could you conduct experiments with large-scale VLA as did in DSRL?"
>
> We conducted a new experiment using the large-scale $\pi_0$ model on the Libero-90 "pick up cream cheese" task over three seeds, comparing against DSRL. LP-DS successfully steers this VLA (approx. 0.9 success rate at step 60000), highlighting that we can effectively adapt high-capacity architectures using a very small perturbation network (a $128 \times 3$ MLP). Results are on the project site.
>
> "Can the authors provide stronger evidence that the KL approximation in Eq. (5) is a good proxy for staying on-manifold for the frozen decoder?"
>
> This constraint is mathematically principled, not merely an empirical proxy. Bounding the squared L2 perturbation acts as a mathematically rigorous, tractable surrogate for bounding the exact expected conditional KL divergence. This natively guarantees the steered queries remain on the data manifold. Please refer to our detailed response to Reviewer Z1TH for the mathematical derivation of this bound.
>
> We hope the new $\pi_0$ evaluations, baseline comparisons, and clarifications regarding the minimal tuning effort for $\delta$ thoroughly address your concerns about our method's scalability and structural advantages. Please let us know if any further clarifications are needed during the discussion period.

---

> > ### Author Rebuttal · Reviewer_3nvu · 2026-04-03
> >
> > I thank the authors for their thorough responses. My concerns have been fully addressed, and I am satisfied with the clarifications provided. I will accordingly raise our score to reflect this.

---

> > > ### Author Response · Authors · 2026-04-08
> > >
> > > We sincerely thank the reviewer for the positive feedback and for recognizing the improvements made during the rebuttal. As all concerns have been fully addressed and will be incorporated into the revised manuscript, alongside enhancements prompted by other reviewers, we kindly invite you to consider if these clarifications merit a further increase in the score toward a full acceptance. We believe the current version significantly strengthens the contribution of the work.

---

### Decision · Program_Chairs · 2026-04-30

**Decision:**

Accept (regular)

**Comment:**

The authors explore the question of how to adapt frozen generative policies with reinforcement learning without destabilizing the pretrained decoder or destroying multimodal behavior. Overall, the submission's major contribution concerns a simple but effective latent-space adaptation framework, LP-DS, which learns a state-conditioned residual perturbation in Gaussian noise space and regularizes it with a Lagrangian trust-region objective so that policy improvement remains close to the backbone prior and avoids off-manifold latent queries. I find the paper strong overall: the method is technically clean, the motivation is clear, the empirical study is broad across RoboMimic, Gym, and Adroit, and the results consistently show improved success, return, and sample efficiency over representative baselines, while the toy study, action-entropy analysis, and trust-region ablations provide useful evidence that the proposed constraint mitigates mode collapse rather than merely improving performance by brute-force optimization. Although the approach still introduces an environment-dependent trust-region hyperparameter and the diversity analysis relies partly on proxy metrics, the paper presents a well-motivated and practically relevant contribution with solid validation. On balance, I support accept.